# Wider or Deeper? Scaling LLM Inference-Time Compute with Adaptive Branching Tree Search

**Yuichi Inoue***, **Kou Misaki***, **Yuki Imajuku, So Kuroki, Taishi Nakamura, Takuya Akiba**

Sakana AI, Japan

{y.inoue,takiba}@sakana.ai

## Abstract

Recent advances demonstrate that increasing inference-time computation can significantly boost the reasoning capabilities of large language models (LLMs). Although repeated sampling (i.e., generating multiple candidate outputs) is a highly effective strategy, it does not leverage external feedback signals for refinement, which are often available in real tasks like coding. In this work, we propose *Adaptive Branching Monte Carlo Tree Search (AB-MCTS)*, a novel inference-time framework that generalizes repeated sampling with principled multi-turn exploration and exploitation. At each node in the search tree, AB-MCTS dynamically decides whether to "go wider" by expanding new candidate responses or "go deeper" by revisiting existing ones based on external feedback signals. We evaluate our method on complex coding and engineering tasks using frontier models. Empirical results show that AB-MCTS outperforms both repeated sampling and standard MCTS, underscoring the importance of combining the response diversity of LLMs with multi-turn solution refinement for effective inference-time scaling. Code is available at: https://github.com/SakanaAI/treequest.

## 1 Introduction

Recent work has shown that *inference-time scaling*, namely allocating more computation at inference time, can markedly boost the performance of large language models (LLMs) on complex tasks. As outlined in Section 2, existing approaches to inference-time scaling fall into three broad categories: *(1)* post-training fine-tuning, *(2)* reward-guided chain-of-thought (CoT) generation, and *(3)* multiple answer generation. In this paper, we focus on the third category. The multiple answer generation approach repeatedly queries an LLM at non-zero temperature to produce a set of candidate outputs and then selects the most promising one. This approach enhances the LLM's problem-solving abilities on-the-fly, without further training [1–11]. Because it is orthogonal to the other two families, it can be seamlessly combined with them.

The most widely successful approach in this category is *repeated sampling*, which includes techniques such as best-of-$n$ sampling, majority voting, and self-consistency [2, 3, 12]. In repeated sampling, an LLM at non-zero temperature generates multiple candidate outputs independently from the same initial prompt, and a final solution is selected, typically by a simple heuristic. This paradigm has proved effective on challenging benchmarks, including coding competitions [1, 3] and ARC-AGI [13]. The strategy leverages the *diverse and vast output space* exposed by LLM generation, and sampling more responses increases the odds that one of them is high-quality. The empirical success of repeated sampling underscores that harnessing this diversity is central to effective inference-time scaling.

However, repeated sampling focuses exclusively on *exploration* and lacks an explicit mechanism for *exploitation*. In certain real-world scenarios, one can obtain external feedback on a candidate

---

*Equal contribution. See author contributions for details.

39th Conference on Neural Information Processing Systems (NeurIPS 2025).

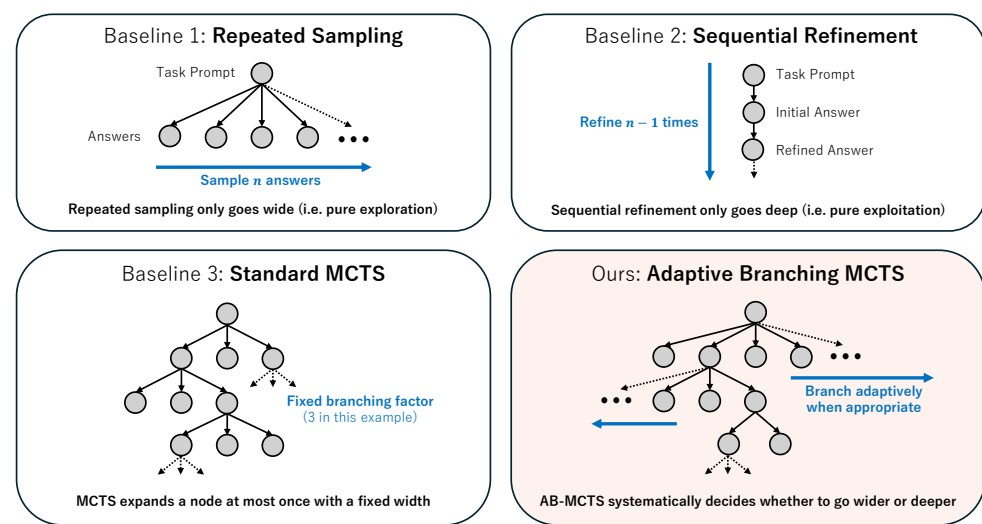

Figure 1: **Visual comparison of AB-MCTS vs. baselines.** Unlike baselines that are purely wide (repeated sampling), purely deep (sequential refinement), or fixed-width (standard MCTS), AB-MCTS dynamically decides whether to branch outward or drill down, unifying both search directions.

solution. For instance, in coding tasks, one can run tests to evaluate the correctness of generated programs and gather feedback on how to improve them [4, 5, 14]. In such settings, it is natural to select promising solutions and refine them based on available feedback, which repeated sampling alone cannot accomplish effectively.

Several approaches [6, 7, 9–11, 15] have been proposed for exploration and exploitation in such multi-turn settings, but the majority were designed before the power of inference-time scaling was fully recognized. Consequently, these methods use a fixed "width", i.e., they treat the number of answers generated from a single prompt as a fixed hyperparameter. For example, methods based on standard Monte Carlo Tree Search (MCTS) use a fixed branching factor (i.e., the number of child nodes per state) as a hyperparameter [9–11, 15]. As demonstrated by the success of repeated sampling, effective inference-time scaling requires leveraging a diverse and vast output space, thus, providing substantial evidence that a fixed width hinders scaling.

In this work, we propose *Adaptive Branching Monte Carlo Tree Search (AB-MCTS)*, a novel inference-time framework that generalizes repeated sampling with multi-turn exploration and exploitation (Figure 1). The main technical challenge is to introduce unbounded branching into MCTS. Unlike traditional MCTS, AB-MCTS does not fix the width as a static hyperparameter. Instead, at each node of the search tree, AB-MCTS adaptively decides whether to explore ("go wider") by generating new candidate responses or exploit ("go deeper") by refining existing ones, leveraging external feedback signals. Under the hood, we formalize our decision process via Bayesian posterior updates, ensuring each expansion balances exploration and exploitation in a principled manner. This design naturally extends repeated sampling, allowing us to harness the diverse and vast output space of LLMs when necessary. Consequently, our framework provides a powerful mechanism for balancing exploration and exploitation in the context of LLM inference-time scaling.

We evaluated AB-MCTS on complex coding and machine learning engineering benchmarks [1, 16], as well as ARC-AGI [17], using frontier models such as GPT-4o [18] and DeepSeek-V3 [19], in a scenario that scales up inference-time compute by allowing multiple generation calls for each task instance. Under the same computational budget, AB-MCTS achieved better results than previous approaches, such as repeated sampling and standard MCTS.

**Contributions.** ① We highlight the challenge of effectively incorporating unbounded branching into tree search. This is pivotal for combining the power of the diverse and vast output space of LLMs, a cornerstone of inference-time scaling, with solution refinement. ② To address this challenge, we introduce AB-MCTS, which systematically decides whether to "go wider" or "go deeper." We present two variants, AB-MCTS-M and AB-MCTS-A, based on different principles, each offering distinct trade-offs. ③ In a practical setting using frontier models and real-world complex tasks, we show that AB-MCTS outperforms existing methods.

## 2  Related Work

**Inference-Time Scaling by Post-Training Fine-Tuning.** Recent post-training work, exemplified by OpenAI o1/o3 [20, 21], uses reinforcement learning or supervised CoT fine-tuning to deepen LLM reasoning and boost single-answer quality [20–25]. Our approach instead generates many candidates and refines them with external feedback, pursuing a complementary objective.

**Inference-Time Scaling via Reward-Guided CoT.** Reward-guided CoT scales inference by searching one step (typically a sentence) at a time [26–34]. Primarily for math tasks, it aims to improve single-answer quality, making it orthogonal to our multiple-answer generation approach.

**Inference-Time Scaling by Multiple Answer Generation.** Since the community has come to appreciate the power of inference-time scaling, the strategy that has been studied widely is repeated sampling, in which the model generates many candidate answers and selects the best one [1–3, 35]. Although empirically strong and widely used, repeated sampling leaves obvious room for improvement because it does not refine its candidates using external feedback [4, 5]. Before the era of large-scale inference-time compute, a variety of task-specific strategies were proposed for relatively small scales; examples include tree expansions directed by LLMs [6] and Bayesian methods [7]. LATS [9], RAP [15], SWE-Search [10], and RepoUnderstander [11] combine LLMs with MCTS, primarily targeting sequential decision making. In this context, nodes represent states and edges represent actions, which may involve interaction with an environment. LATS utilizes API calls and code execution as actions to solve tasks. RAP addresses the process of solving block-moving puzzles and mathematical word problems step-by-step. SWE-Search explores sequences of actions such as searching, editing, and running tests to resolve issues within a software repository. RepoUnderstander employs MCTS for exploration on a repository knowledge graph. The application of LATS to coding tasks [9, Section 5.2] aligns with the context of multiple-answer generation in this paper and corresponds to what we refer to as "standard MCTS" in our experiments.

**Progressive Widening in MCTS.** Progressive widening (PW) [36, 37] is a classic technique that gradually increases actions considered per node. It was designed for games with unique actions and no side information for untried moves, relying on visit-count heuristics. Complementary to PW, Sokota et al. [38] propose "abstraction refining", which groups similar successors using a decreasing similarity threshold and shows advantages over PW under equal simulation budgets in stochastic domains. Our approach differs as new branches are sampled from the same LLM. This homogeneity in generation allows a principled statistical rule for choosing between widening and deepening.

## 3  Method

### 3.1  Preliminaries

First, we introduce the setup and notation, with detailed elaboration provided in Appendix A.1. We consider a setting where an LLM, represented by a function $f_{\text{LLM}}$, receives a textual prompt $t_{\text{in}}$ containing (1) task instructions with optional few-shot examples, and/or (2) previously generated outputs along with external feedback, and generates an answer $t_{\text{out}} = f_{\text{LLM}}(t_{\text{in}})$. A scoring function $R$ then evaluates an answer $t_{\text{out}}$ to produce a score $r = R(t_{\text{out}})$, where higher scores indicate better performance. We typically assume that the score $r$ is normalized to the range $[0, 1]$, but our framework allows for arbitrary ranges as well. Our goal is to find an output $t_{\text{out}}$ that attains a high score $r$ under limited calls to the LLM at inference time. Such tasks arise, for example, in code generation, where the correctness or quality of the output can be quantified; for instance, $R$ may execute the generated code and return the fraction of test cases passed. In some cases, the true score evaluator may be inaccessible (e.g., hidden test cases), so we assume we have access to some surrogate or partial evaluator $R$, such as a public test evaluator, during the search. We aim to leverage this evaluator to guide an efficient search for better solutions.

### 3.2  Adaptive Branching MCTS

**MCTS for LLM-based Answer Generation.** We perform the answer search by constructing a search tree $T$, in which each non-root node $N$ is associated with an LLM-generated answer to a given task. Our goal is to construct $T$ so that it contains answers with scores as high as possible.

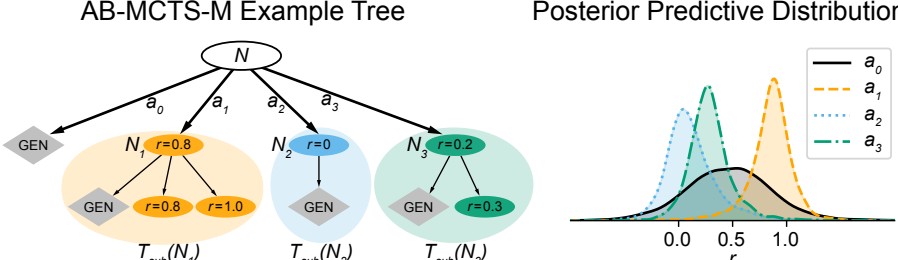

Figure 2: **Example tree structure and score posterior predictive distributions for AB-MCTS with mixed models (AB-MCTS-M).** Here, $a_1$ leads to a set of child nodes with higher scores, causing a peak at larger $r$. As more child samples are collected, the variance of the distribution decreases.

---

**Algorithm 1** Adaptive Branching MCTS

1: **function** AB-MCTS($n_{\text{nodes}}$)
2:   $T \leftarrow$ INITIALIZETREE( )
3:   **for** $n = 1, \ldots, n_{\text{nodes}}$ **do**
4:    $N \leftarrow$ SELECTEXPANSIONTARGET($T$)       ▷ *Step 1. Select an expansion target*
5:    $N_{\text{new}} \leftarrow$ EXPAND($N, T$)      ▷ *Step 2. Expand the selected node to generate a child*
6:    SCOREBACKUP($N_{\text{new}}, T$)       ▷ *Step 3. Backup the score from the generated node*
7:   **return** SELECTBEST($T$)

8: **function** SELECTEXPANSIONTARGET($T$)
9:   $N \leftarrow$ GETROOT($T$)
10:   **while not** ISLEAF($N$) **do**
11:    $N_{\text{next}} \leftarrow$ SELECTCHILD($N, T$)       ▷ *Detailed in Sections 3.3 and 3.4*
12:    **if** ISGENNODE($N_{\text{next}}$) **then**     ▷ *If a GEN node is selected, branch off from the node*
13:     **break**
14:    $N \leftarrow N_{\text{next}}$
15:   **return** $N$

---

For this purpose, we employ MCTS, formulated iteratively as follows. Starting from a single root node, we perform $n_{\text{nodes}}$ iterations, each adding one new node, resulting in a total of $1 + n_{\text{nodes}}$ nodes. Each iteration has three steps: **(1) Selection**, where we select a node $N$ for expansion; **(2) Expansion**, where we expand $N$ by generating a new answer from node $N$, creating a new child node $N_{\text{new}}$ and appending it to $N$. Specifically, if $N$ is the root, the new answer is directly generated from the task prompt; if $N$ is non-root, the new answer refines the answer associated with $N$, using external feedback; and **(3) Score backup**, where we propagate the score of $N_{\text{new}}$ up toward the root of the tree $T$. We adopt different backup rules in our proposed methods (see Section 3.3 and 3.4). In our setting, no separate rollout is needed, since each node's score $r$ can be evaluated directly once an output is generated. After $n_{\text{nodes}}$ iterations, we select the best node based on a chosen criterion.

In standard MCTS, only leaf nodes are selected and expanded, (i.e., each node is expanded at most once), and the expansion adds a fixed number of child nodes. However, since each query to an LLM at non-zero temperature can yield different outputs from the same prompt, the branching factor is theoretically infinite. To accommodate such unbounded branching, we relax the standard MCTS constraints and allow selection and expansion of non-leaf nodes. Moreover, recent studies [3] suggest that drawing many outputs from the same prompt at non-zero temperature can improve performance. Allowing unbounded branching enables us to fully exploit these varied samples, whereas restricting the branching factor could miss correct answer generations and undermine overall performance.

**Adaptive Branching via the GEN Node.** To fully leverage the potential performance improvement from unbounded branching, we allow nodes that have already been expanded once to be expanded again and further branched, unlike in standard MCTS. To explicitly represent the action of generating new child nodes, we introduce a *GEN node*. Every node $N$ (including newly expanded ones during iterations) has a GEN node as a child. When the GEN node with parent node $N$ is selected during the selection step, we expand $N$ by adding a new child node. Algorithm 1 outlines this approach, called *Adaptive Branching Monte Carlo Tree Search* (*AB-MCTS*).

The only remaining component we need is a selection policy, including when to select a GEN node. We propose two algorithms with different selection policies: *AB-MCTS-M* (Mixed model) and *AB-MCTS-A* (node Aggregation). Both follow the overall procedure in Algorithm 1 and use Thompson sampling to balance exploration and exploitation.

The UCT score is inapplicable to our AB-MCTS because GEN nodes make the problem fundamentally different from a standard multi-armed bandit problem, for which UCT was designed. In standard MCTS, the arms (branches) are static. In contrast, the GEN node in AB-MCTS dynamically generates new arms. This special problem setting, where arms are generated on the fly, prevents the direct application of UCT. We, therefore, adopt a Bayesian probabilistic model. This enables Thompson sampling based on the posterior distribution and obviates the need for complex UCB-style confidence bound analysis.

**Thompson Sampling for Node Selection.** In our proposed methods, we employ a Bayesian approach with Thompson sampling for node selection. Here, we employ Thompson sampling because GEN nodes do not have child nodes, making it impossible to compute their UCT scores. In addition, Thompson sampling has the advantage of allowing node expansion in parallel. This is particularly beneficial when evaluating node scores is time-consuming, as in the case of MLE-Bench (See Appendix B.1 for MLE-Bench details).

Concretely, during `SelectChild` step at line 11 of Algorithm 1, we employ Thompson sampling to decide between expanding a GEN node or selecting from existing child nodes at node $N$. Let $N$ be a node with potential actions

$$A_N = \{a_0, a_1, \ldots, a_{n_{\text{child}}}\},$$

where the action $a_0$ corresponds to choosing the GEN node, and $a_1, \ldots, a_{n_{\text{child}}}$ correspond to choosing the already-existing child nodes. Suppose $P_N(r \mid a_i)$ is the posterior predictive distribution of the score $r$ for an eventually expanded new node ($N_{\text{new}}$ at line 5 of Algorithm 1) if we choose the action $a_i$ at node $N$. Then Thompson sampling proceeds by,

1. Calculate $P_N(r \mid a_j)$ for each action $a_j$ at node $N$.
2. Draw scores $r_{N_{\text{new}},a_j}$ from $P_N(r \mid a_j)$ for each action $a_j$.
3. Select $\hat{a} = \arg\max_{a_j \in A_N} r_{N_{\text{new}},a_j}$.

This three-step process corresponds to a single call to `SelectChild`.

A key question is how to perform step 1, i.e., how to model and calculate $P_N(r \mid a_j)$ for all $a_j$, in particular for $j = 0$ (i.e., GEN node). We address this with two strategies: a mixed Bayesian model (*AB-MCTS-M*) and a node aggregation method (*AB-MCTS-A*). In both cases, we model the score probability distributions by Bayesian posterior predictives, but with different statistical models.

### 3.3 AB-MCTS-M: Adaptive Branching MCTS with Mixed Model

To model $P_N(r \mid a_j)$, we employ a node-specific mixed model fitted individually at each node $N$. That is, we fit a separate model for each $N$ every time `SelectChild` in Algorithm 1 is invoked. Denoting $r_{N_{\text{new}},a_j} \sim P_N(r \mid a_j)$ as a score of an eventually expanded node $N_{\text{new}}$ if we choose an action $a_j$ at $N$, our mixed model is given as:

$$
\begin{aligned}
r_{N_{\text{new}},a_j} = \alpha_j + \sigma_y \epsilon_{N_{\text{new}}}, \quad \alpha_j = \mu_\alpha + \sigma_\alpha \epsilon_j, \\
\epsilon_{N_{\text{new}}} \sim \mathcal{N}(0,1), \quad \epsilon_j \sim \mathcal{N}(0,1),
\end{aligned}
\tag{1}
$$

Here, $\alpha_j$ is a "group-level" intercept capturing the quality of the base solution at $N_j$, while $\sigma_y \epsilon_{N_{\text{new}}}$ represents per-instance noise. To fit this model, we place priors on the hyperparameters ($\mu_\alpha, \sigma_\alpha, \sigma_y$) and employ Markov Chain Monte Carlo (MCMC) to sample from their posterior distribution. The GEN node (action $a_0$) is treated as a newly introduced group without its own direct observations. However, its group-level intercept $\alpha_0$ is inferred not from the prior alone but rather from the posterior distribution over $\mu_\alpha$ and $\sigma_\alpha$, which is informed by the other observed data. We assume that even after multiple refinement stages, the quality associated with the answer at node $N_j$ continues to be captured by this shared parameter (see Appendix A.3.2 for further details).

**Algorithm Outline.** To model $P_N(r \mid a_j)$, AB-MCTS-M assigns each subtree under $N_j$, denoted as $T_{\text{sub}}(N_j)$, as a distinct group $j$ (see Figure 2 for example subtree). The mixed model leverages

observed scores from these groups to compute the posterior predictive distributions of expected scores for new nodes generated from each group (See Figure 2 for a schematic illustration). We sample the scores from all the groups (the GEN node and $T_{\text{sub}}(N_j)$) using calculated posterior predictives. If the GEN node's sampled score is highest, we call $f_{\text{LLM}}$ to generate a new child node. Otherwise, we choose the child node $N_j$ with the highest score and continue the sampling step.

**Score Backup Mechanism.** When a new node $N$ is created, its observed score is added to the histories of $N$ and its ancestors. This cumulative record is used to update the posterior distributions in the mixed model. The observed score is not backed up to a GEN node, but it indirectly influences the GEN node's score probability distribution through the shared parameters in the mixed model (see Appendix A.5 for a detailed walkthrough).

### 3.4 AB-MCTS-A: Adaptive Branching MCTS with Node Aggregation

In AB-MCTS-M, during the selection step at $N$, we use a mixed model that shares statistical strength across groups through the shared model parameters. In contrast, AB-MCTS-A is designed in the same spirit as the standard UCT-based MCTS, and there are no shared model parameters among the different actions. This design simplifies the statistical modeling and makes the computation more lightweight compared to AB-MCTS-M.

The major problem is how to back up scores to GEN nodes. Since the generated node is not attached as a child to a GEN node, it makes the backup of scores difficult to define. Here, we introduce a *CONT node* at the same tree level as all the GEN nodes (see Figure 3). Intuitively, the CONT node represents the action of continuing refinement from the current answer at node $N$, rather than generating a new node. By explicitly separating these two actions–generating new answers (GEN) and refining existing answers (CONT)–we create a clear path for score propagation. Specifically, the score of the expanded node is first backed up to the GEN node, and since all ancestors of the GEN node are either nodes with LLM answers or CONT nodes, the score subsequently does not propagate through other GEN nodes (see Figure 3 for an example tree).

**Algorithm Outline.** AB-MCTS-A aggregates all child nodes under a single CONT node, which represents refinements from existing child nodes (see Figure 3). We model each node's score probability in a Bayesian framework and perform Thompson sampling on posterior predictives to decide between generating a new child (GEN) or refining an existing one (CONT). In contrast to AB-MCTS-M, we do not use shared parameters among different node probability distributions.

AB-MCTS-A Example Tree

Figure 3: **Example tree structure for AB-MCTS-A.** All child nodes are aggregated under a CONT node, and a GEN node doesn't have child nodes.

To model $P_N(r \mid a_j)$, we utilize exponential family distributions with conjugate priors, enabling analytical and efficient posterior updates. We employ two variants:

1. AB-MCTS-A (Gaussian), using a normal-inverse-$\chi^2$ prior for unbounded scores: $P_N(r \mid a_j) = p(r \mid \{r_k\}_{k=1}^K) = \mathcal{N}(r \mid \hat{m}, \frac{\sigma^2}{\hat{\kappa}})\chi^{-2}(\sigma^2 \mid \hat{\nu}, \hat{\tau}^2)$, and

2. AB-MCTS-A (Beta), using a Beta prior for scores in $[0, 1]$: $P_N(r \mid a_j) = p(r \mid \{r_k\}_{k=1}^K) = B(r \mid \hat{\alpha}, \hat{\beta})$,

where $r_k$ represents the scores backed up to the node $N_j$ (GEN node, CONT node or LLM-generated child nodes; see Figure 3 for example tree), where $\hat{m}, \hat{\kappa}, \hat{\nu}, \hat{\tau}, \hat{\alpha}, \hat{\beta}$ are determined from observed scores $r_k$ and updated as these scores are backed up. The detailed parameter update rules are given in Appendix A.4.

**Score Backup Mechanism.** During score-backup operations, the expanded node score is backed up to the GEN node which led to the expansion of that node and the GEN node's ancestors (see Appendix A.5 for a detailed walkthrough). As we can see from Figure 3, a GEN node's ancestors include only generated nodes and CONT nodes, so the score is backed up to a GEN node only from

Table 1: **Performance of AB-MCTS against baselines across benchmarks and models.** This table compares AB-MCTS with the baseline methods. Evaluations were performed on LiveCodeBench, CodeContest, and ARC-AGI using GPT-4o and DeepSeek-V3 with a maximum generation budget ($2^7$). Each entry provides a performance score (higher values are better) and its corresponding rank (in parentheses, 1st is best). The "Avg. Rank" column shows the average rank across all settings.

| Method | LiveCodeBench | | CodeContest | | ARC-AGI | | Avg. Rank |
|---|---|---|---|---|---|---|---|
| | GPT-4o | DeepSeek-V3 | GPT-4o | DeepSeek-V3 | GPT-4o | DeepSeek-V3 | |
| Repeated Sampling | $37.8 \pm 0.5$ (4) | $40.7 \pm 1.9$ (6) | $37.9 \pm 0.3$ (4) | $43.2 \pm 0.9$ (5) | **$15.0 \pm 1.0$ (1)** | **$18.6 \pm 1.0$ (1)** | 3.5 |
| Sequential Refinement | $37.8 \pm 2.4$ (4) | $41.6 \pm 0.6$ (5) | $30.1 \pm 0.3$ (6) | $41.6 \pm 0.9$ (6) | $8.7 \pm 0.9$ (6) | $10.0 \pm 0.6$ (6) | 5.5 |
| Standard MCTS | $36.7 \pm 1.0$ (6) | **$43.2 \pm 2.1$ (1)** | $37.5 \pm 0.0$ (5) | $43.8 \pm 0.9$ (3) | $9.0 \pm 1.5$ (5) | $14.0 \pm 1.5$ (5) | 4.2 |
| AB-MCTS-M | $38.9 \pm 1.9$ (2) | $43.0 \pm 1.5$ (2) | **$40.6 \pm 1.0$ (1)** | $44.6 \pm 0.9$ (2) | $12.3 \pm 1.2$ (4) | $16.0 \pm 1.0$ (3) | **2.3** |
| AB-MCTS-A (Gaussian) | **$39.1 \pm 1.9$ (1)** | $42.5 \pm 1.5$ (3) | $40.2 \pm 1.7$ (3) | $43.4 \pm 0.9$ (4) | $13.0 \pm 3.6$ (3) | $18.3 \pm 0.6$ (2) | 2.7 |
| AB-MCTS-A (Beta) | $38.7 \pm 1.2$ (3) | $42.3 \pm 0.8$ (4) | $40.4 \pm 0.3$ (2) | **$44.8 \pm 0.6$ (1)** | $14.0 \pm 2.1$ (2) | $16.6 \pm 0.6$ (4) | 2.7 |

the node that is created by choosing that GEN node. The backed-up score is used to update prior probability distribution parameters.

# 4 Experiments

## 4.1 Experimental Setup

**Benchmarks.** We evaluated AB-MCTS on four diverse benchmarks that require complex problem-solving: LiveCodeBench [14], CodeContest [1], ARC-AGI [17], and MLE-Bench [16]. Live-CodeBench and CodeContest consist of competitive programming problems that demand mathematical and algorithmic reasoning. ARC-AGI involves abstracting a common transformation rule from visual patterns and implementing it as code. MLE-Bench, derived from Kaggle competitions, involves constructing and optimizing machine learning models to achieve high scores based on the evaluation metrics of each competition. For all these benchmarks, LLMs generate Python code to solve each task, and external feedback (e.g., test case results, validation scores) is available to guide the search. More details on the benchmarks can be found in Appendix B.1.

**Models.** We perform our experiments using GPT-4o (`gpt-4o-2024-08-06`) [18], and DeepSeek-V3 (`deepseek-chat`) [19]. Each LLM generates a complete solution in a single API call. We define the generation budget as the maximum number of API calls and set it to $2^7 = 128$. The temperature was set to 0.6 for GPT-4o following [3] and 1.0 for DeepSeek-V3 following the official documentation.

**Baselines.** We benchmark AB-MCTS against three representative approaches. **(1) Repeated Sampling (Best-of-$n$)** [1, 3, 39] independently generates up to $n$ candidate solutions from a single LLM prompt, a simple yet competitive baseline for coding tasks. **(2) Sequential Refinement** [4] iteratively improves each solution by re-prompting the LLM with its own output and feedback. **(3) Standard MCTS** follows the configuration from LATS [9, Section 5.2] (See also Appendix C.8). Each expansion adds five child nodes, and the search proceeds until it reaches the $2^7$ nodes, with the final expansion creating only three nodes to meet this limit precisely. Hyper-parameters for AB-MCTS are summarized in Appendix B.2.

## 4.2 Results

As detailed in Tables 1 and 2, our comprehensive evaluations reveal AB-MCTS as a consistently superior approach across diverse benchmarks and LLMs, achieving the top average rank and outperforming established baselines. This consistent success stems from the distinctive ability of AB-

Table 2: **Performance on MLE-Bench tasks.** AB-MCTS-M demonstrates robust performance, achieving the best average rank across diverse ML tasks.

| Method | Nomad2018 | Spooky. | Pizza. | Avg. |
|---|---|---|---|---|
| Repeated Sampling | 0.065 (3) | 0.47 (4) | 0.72 (2) | 3.0 |
| Sequential Refinement | **0.059 (1)** | 0.46 (3) | 0.62 (3) | 2.3 |
| Standard MCTS | 0.076 (4) | 0.45 (2) | 0.60 (4) | 3.3 |
| AB-MCTS-M | 0.060 (2) | **0.38 (1)** | **0.72 (1)** | **1.3** |

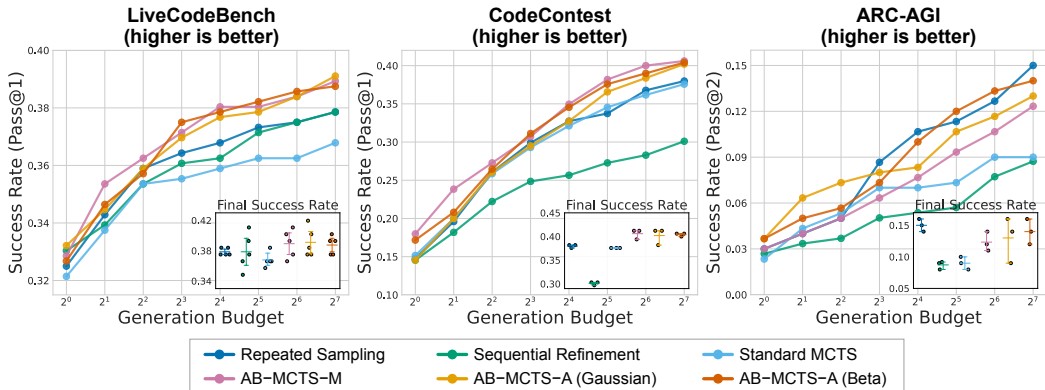

Figure 4: **Performance comparison on LiveCodeBench, CodeContest, and ARC-AGI.** We compare the six methods using GPT-4o by plotting the success rate against the generation budget. The inset plots provide a detailed view of performance at a maximum generation budget ($2^7$); the mean success rate, its 95% confidence interval, and the results from the individual runs are shown. Variance at a generation budget of $2^0$ arises from conducting each experiment independently with nonzero temperature. See Figure 6 for experiments on ARC-AGI with a larger budget.

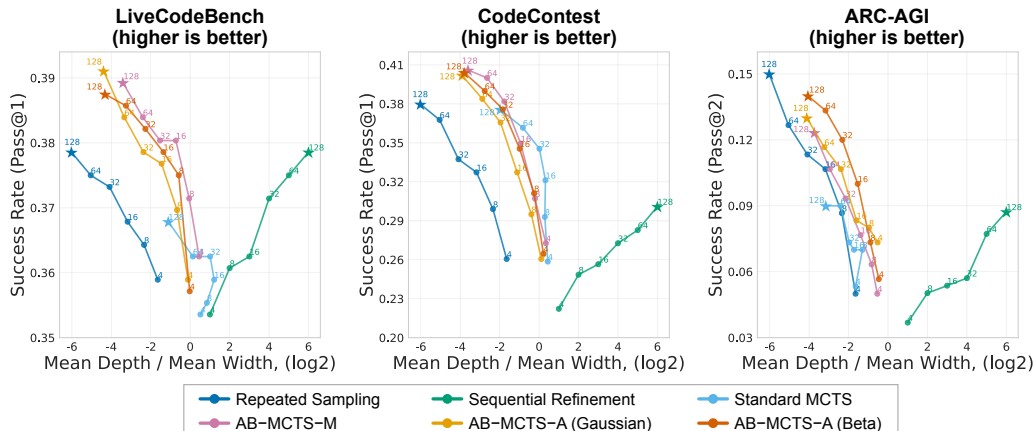

Figure 5: **Comparing algorithms by search tree shape and performance.** Each point shows the performance against the average tree shape for a given algorithm at a specific generation budget. The x-axis represents the log-ratio of mean depth to mean width. Mean width is the average number of nodes per depth. Larger and smaller x-axis values indicate deeper and wider searches, respectively.

MCTS to dynamically adapt its search strategy by precisely balancing exploration and exploitation to the varying demands of each problem, an adaptability largely absent in baseline methods. We next detail these results by benchmark, followed by an analysis of the search behavior of AB-MCTS.

**LiveCodeBench and CodeContest.** Figure 4 (left and center) reports the success rate (Pass@1) versus the generation budget for GPT-4o on LiveCodeBench and CodeContest. As expected, all methods demonstrate improved performance with increasing computational budget. On both benchmarks, AB-MCTS algorithms generally outperform the baseline methods. Notably, on LiveCodeBench (Figure 4 left), AB-MCTS starts to pull ahead of the baselines even with a small budget of $2^3$. On CodeContest (Figure 4 center), AB-MCTS demonstrates superior performance compared to baselines at larger budgets of $2^5$ and beyond. Appendix Figure 8 shows that while standard MCTS performs relatively well with DeepSeek-V3 compared to GPT-4o, our proposed methods achieve a comparable success rate on LiveCodeBench and surpass standard MCTS on CodeContest.

**ARC-AGI.** Figure 4 (right) shows the performance on ARC-AGI, a particularly challenging benchmark. Following ARC-AGI's official evaluation protocol, we report Pass@2 (Pass@1 is also reported in Appendix C.6). Consistent with previous work [13], repeated sampling proves to be a strong baseline in our setup, indicating the importance of broad exploration for this task. While standard

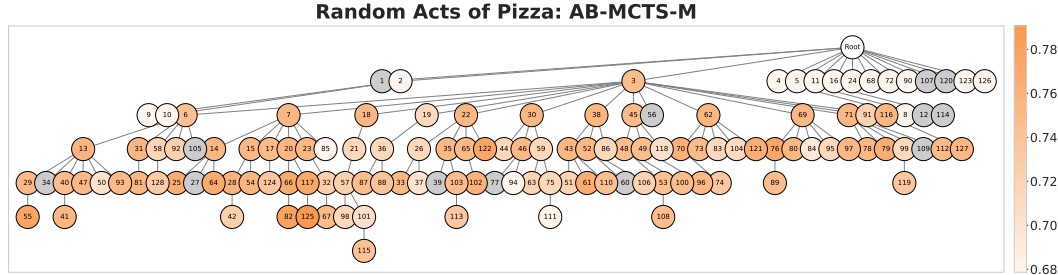

Figure 7: **Example search tree generated by AB-MCTS-M on Random Acts of Pizza (MLE-Bench).** The figure shows how AB-MCTS-M dynamically balances exploration and exploitation. Each node represents a solution. Nodes are colored according to their evaluation score used as the search signal for AB-MCTS-M. The number inside each node indicates the order of generation. Grey nodes mark candidates whose code failed to execute and therefore received no score.

MCTS yields only marginal improvements with larger budgets, our AB-MCTS framework achieves performance comparable to repeated sampling. This suggests AB-MCTS's capability to effectively explore potentially by dynamically widening its search when beneficial. Similar results were observed with DeepSeek-V3, as detailed in Appendix Figure 8.

**MLE-Bench.** Table 2 and Appendix Figure 10 present the performance on three competitions from MLE-Bench using GPT-4o. Since MLE-Bench requires substantial GPU resources for training and evaluating machine learning models, we exclusively used GPT-4o and focused on the baseline methods and AB-MCTS-M (see also Appendix B.1). The best-performing baseline method varies across competitions. This again highlights that different tasks benefit from different exploration-exploitation trade-offs. In contrast, AB-MCTS-M consistently delivers strong performance in these tasks. This consistent success across diverse competitions underscores AB-MCTS-M's inherent strength in effectively adapting its search strategy to varying problem structures.

## 4.3 Analysis

**Analysis of Search Behavior: Width vs. Depth.** To quantitatively analyze how AB-MCTS balances exploration and exploitation, we examined the average depth and the average width at each depth of the generated search trees. Figure 5 shows that AB-MCTS methods tend to generate wider trees compared to standard MCTS. This occurs because AB-MCTS can adaptively decide to explore wider (select the GEN node) from any existing node, unlike standard MCTS. This mechanism allows for more flexible exploration across various tree depths (See also Appendix C.7). In addition to this flexibility in exploring wider, as seen in Table 2, AB-MCTS also achieves strong performance on benchmarks where sequential refinement excels, suggesting that AB-MCTS effectively identifies and exploits promising branches by selecting existing child nodes for refinement. This adaptive nature allows it to combine the strengths of exploration and exploitation, resulting in robust performance across diverse benchmarks.

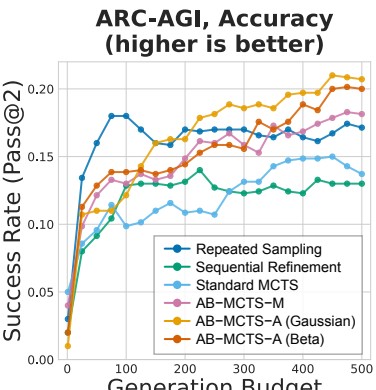

Figure 6: **Performance comparison on ARC-AGI with increased budget.** Scalability of AB-MCTS was assessed with a generation budget extended up to 512. Plotted points represent moving averages to clarify performance trends.

**Scaling with Increased Budget.** Highly complex problems often require a substantial generation budget to find a correct solution. ARC-AGI is a prime example, where extensive exploration via repeated sampling is known to improve performance even at large budgets as reported in [13]. To investigate the scaling properties of our approach, we extended the experiments on ARC-AGI using DeepSeek-V3 with a larger generation budget up to $2^9 = 512$. As shown in Figure 6, the performance

of AB-MCTS continues to improve substantially as the budget increases from 200 to 500, while the improvement rate of repeated sampling begins to plateau. Standard MCTS also continues to improve with a larger budget, yet shows a significantly lower success rate compared to the AB-MCTS methods. This performance gap highlights that AB-MCTS is more effective at directing its search towards promising branches within the search tree at large computational scales.

**Qualitative Analysis of the Search Trees.** Figure 7 and Appendix Figure 11 present example search trees generated by AB-MCTS-M and standard MCTS. These visualizations illustrate more adaptive branching by AB-MCTS-M compared to standard MCTS. This adaptive nature reveals that AB-MCTS-M flexibly balances exploration and exploitation throughout the search process, dynamically allocating budget to explore diverse new candidates ("going wider") and refine promising ones ("going deeper"). Further discussion can be found in Appendix C.3.

**Efficiency and Performance against Repeated Sampling.** While repeated sampling benefits from potential efficiencies such as parallel sampling and no feedback computation costs, our results demonstrate the significant advantages of AB-MCTS. On ARC-AGI, where repeated sampling is notably strong, AB-MCTS not only continues to improve with increased budget but also ultimately achieves performance levels that repeated sampling cannot reach (Figure 6). Furthermore, on LiveCodeBench and CodeContest, AB-MCTS variants can reach the peak performance of repeated sampling substantially earlier in many cases (Figure 4, Appendix Figure 8). This indicates that even when accounting for the inherent advantages of repeated sampling, AB-MCTS emerges as a promising approach to efficiently use the generation budget to achieve superior results in diverse scenarios.

## 5 Conclusions

This paper introduced Adaptive Branching Monte Carlo Tree Search (AB-MCTS), a novel inference-time framework to enhance LLM performance on complex tasks by effectively integrating multi-turn exploration and exploitation. Unlike previous methods, AB-MCTS dynamically decides to "go wider" or "go deeper" based on external feedback, leveraging Bayesian decision-making. Our experimental results show AB-MCTS outperforms repeated sampling and standard MCTS, demonstrating the value of adaptively handling the challenge of unbounded branching for effective inference-time scaling.

**Limitations.** Our approach assumes the existence of a reliable score evaluator, but developing such an evaluator itself can be challenging depending on the task. Future work could also explore search strategies that incorporate more fine-grained real-world cost factors beyond API call counts, potentially enhancing the practical utility of AB-MCTS. We believe that addressing these challenges will further enhance the applicability of AB-MCTS across a wider range of problems.

## Author Contributions

Kou Misaki co-designed AB-MCTS and implemented its algorithm and core experimental code. Yuichi Inoue designed and led the experiments, proposed and conducted the experimental analysis, and co-led experimental code development. Yuki Imajuku conducted the CodeContest and Live-CodeBench experiments and co-led experimental code development. So Kuroki implemented and conducted the MLE-Bench experiments. Taishi Nakamura conducted the ARC-AGI experiments. Takuya Akiba initiated the project, co-designed AB-MCTS, advised on experimental code design, and provided overall supervision. All authors contributed to experimental code development, results interpretation, and manuscript refinement.

## Acknowledgements

The authors would like to thank Edoardo Cetin, Luke Darlow, Taro Makino, Kosuke Nakago, Makoto Shing, and Yutaro Yamada for helpful feedback on an earlier version of the draft.

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

# A  Method Details

## A.1  Extended Preliminaries

### A.1.1  Problem Setup

First, we define the problem setting and introduce the mathematical notation.

We consider problems where the input is a natural language prompt $t_{\text{in}}$ (which may include few-shot examples, task instructions, etc.). Given $t_{\text{in}}$, an LLM produces a natural language output $t_{\text{out}}$, which is then scored by a score evaluator to yield a final score $r$. We assume $0 \leq r \leq 1$, where higher values of $r$ correspond to better answers.

This two-stage pipeline can be expressed as:

$$r = R(t_{\text{out}}) = R(f_{\text{LLM}}(t_{\text{in}})), \tag{2}$$

where the function $f_{\text{LLM}}$ represents an LLM that generates the answer, and $R$ is the score evaluator. $f_{\text{LLM}}$ is stochastic and may produce different outputs $t_{\text{out}}$ for the same input $t_{\text{in}}$. Here we allow $t_{\text{out}}$ to include information beyond the direct answer (e.g., the reasoning steps), and we assume that $R$ properly performs the parsing of this answer to perform the score evaluation.

Our framework applies to any task whose final answer can be quantitatively scored, such as programming tasks [1, 14], math problems [29], and machine learning competitions [16]. We assume the score evaluator $R$ is already defined in a task-specific manner, and our goal is to find $t_{\text{out}}$ that attains as high an $r$ as possible during the inference-time answer search.

In some tasks that mimic real competitions, the ground-truth score evaluator $R_{\text{gt}}$ (reflecting the correctness of the answer) is not accessible during the answer search stage. For example, in MLE-Bench [16], the test dataset used for final evaluation is withheld, and in coding competition tasks [1, 14], participants can often only submit their code a limited number of times to obtain the score on hidden test cases. In such situations, to search for the best answer, one may resort to a different score evaluator. For instance, in MLE-Bench, this could be the performance on a public dataset, while in coding competitions, it might be the fraction of solved public test cases. For mathematical tasks, a separately trained reward model [40] may be used. Throughout this work, we assume there is some accessible score evaluator at the answer search stage that can assess the quality of $t_{\text{out}}$.

### A.1.2  Existing Methods for Inference-Time Answer Search

We now review two standard approaches that focus on exploration or exploitation alone.

**Only Go Wide: Repeated Sampling.** A straightforward approach to inference-time answer search is to repeatedly sample an answer from the LLM with a nonzero temperature. We refer to each sampling step as the *direct answer generation process*. By performing it $n$ times,

$$t_{\text{out}}^m = f_{\text{LLM}}(t_{\text{in}}), \quad m \in \{1, \dots, n\}, \tag{3}$$

we obtain multiple candidate answers. Then, one can select the best answer based on a predefined criterion, such as the highest score $r$ (best-of-$n$), majority voting, or self-consistency. Brown et al. [3] recently showed that as $n$ increases, the coverage of generated answers improves. A similar approach was employed by AlphaCode [1] to achieve human-level performance on competitive programming tasks.

**Only Go Deep: Sequential Refinement.** Alternatively, we can leverage the answer refinement process and apply it sequentially to perform the answer search. We consider the situation where we already let some answer generator solve the problem at hand $k$ times, and collected the input-output pairs $t_{\text{in}}^j$ and $t_{\text{out}}^j$ where $j \in \{1, \dots, k\}$ for those answer generations.

We define the *answer refinement process* as a two-step procedure: (1) creation of a new refinement input from the existing input-output pairs, and (2) generation of a new answer $t_{\text{out}}^{k+1}$ from $t_{\text{in}}^{k+1}$. Symbolically,

$$t_{\text{out}}^{k+1} = f_{\text{LLM}}(t_{\text{in}}^{k+1}) = f_{\text{LLM}}\left(h_{\text{refine}}\left(\{t_{\text{in}}^j, t_{\text{out}}^j\}_{j \in \{1,\dots,k\}}\right)\right), \tag{4}$$

where $h_{\text{refine}}$ is a refinement input generator that provides all the information necessary for refinement, such as feedback on each answer (e.g., code execution results or errors for coding tasks).

Applying this refinement step iteratively yields $n$ answers:

$$t_{\text{in}}^1 = t_{\text{in}}, \tag{5}$$

$$t_{\text{out}}^1 = f_{\text{LLM}}(t_{\text{in}}^1), \tag{6}$$

$$t_{\text{in}}^m = h_{\text{refine}}\left(\{t_{\text{in}}^j, t_{\text{out}}^j\}_{j \in \{1,\dots,m-1\}}\right) \tag{7}$$

$$t_{\text{out}}^m = f_{\text{LLM}}(t_{\text{in}}^m) \tag{8}$$

for $m \in \{2, \dots, n\}$. Finally, one selects the best candidate among $\{t_{\text{out}}^a\}_{a \in \{1,\dots,n\}}$ based on a chosen criterion, similar to the repeated sampling approach.

We can regard the two methods above (pure exploration and pure exploitation) as special cases of a tree search: the former expands only from the root node, while the latter continues from the most recently reached leaf node on a single linear path, exploring it in depth without branching outward. The standard MCTS naturally incorporates the two, while it uses a fixed branching factor, thereby limiting the performance gain from the repeated sampling when using LLM. Our AB-MCTS employs a more flexible branching algorithm and effectively leverages performance improvements obtained by repeated sampling.

## A.2 Comparison of AB-MCTS and Standard MCTS with Progressive Widening

The progressive widening has parameters $(k, \alpha)$ which bounds the number of branching factors by $kn^\alpha$ with node visit count $n$. With these parameters, the rule for whether to branch is pre-determined as a function of the node's visit count. Crucially, this decision does not use important information gathered during the search, namely, the observed rewards of the expanded nodes. The UCT score is only used to select which child to descend to after the decision has been made not to branch. Furthermore, defining the branching rule as a function of visit count means that the scaling behavior of the tree's shape and node degrees is pre-determined by the choice of hyperparameters.

In contrast, our approach does not restrict the branching rule based solely on visit counts and hyperparameters. Instead, the branching factor adapts dynamically based on the observed rewards. This is an important requirement for LLM test-time inference scaling, where a tree search that purely goes wide is known to be a strong baseline. To demonstrate the robustness of AB-MCTS, we conducted an experiment to compare AB-MCTS and progressive widening in Appendix C.4.

## A.3 AB-MCTS-M Details

### A.3.1 Background on Mixed Models

Mixed linear models are extensions of traditional linear models that explicitly model non-independence among observations by incorporating both fixed and random effects. Fixed effects capture consistent, predictable patterns across the entire population or dataset (e.g., treatment effects common to all groups). In contrast, the random effects model variability arises within or between specific nested or hierarchical groups (e.g., individual differences among participants, or variations between schools).

### A.3.2 Detailed Mixed Model Formulation and Example Code

In AB-MCTS-M, we fit a separate mixed model at each node $N$ of the MCTS tree, that is, at each sub-step within the MCTS selection step. Specifically, let $N_j$ ($j = 1, \dots, n_{\text{child}}$) denote the direct child nodes of $N$, and define $T_{\text{sub}}(N_j)$ as the subtree under $N_j$, including $N_j$ itself. For a newly generated node $\tilde{N}$ where (i) $j = 0$ (i.e. for GEN node) and $\tilde{N}$ is a direct child of $N$, or (ii) $j = 1, \dots, n_{\text{child}}$ and $\tilde{N}$ is a node expanded from some node in $T_{\text{sub}}(N_j)$ at that iteration step, we assume:

$$r_{\tilde{N}} = \alpha_j + \sigma_y \epsilon_{\tilde{N}}, \quad \alpha_j = \mu_\alpha + \sigma_\alpha \epsilon_j, \tag{9}$$

$$\epsilon_{\tilde{N}} \sim \mathcal{N}(0, 1), \quad \epsilon_j \sim \mathcal{N}(0, 1), \tag{10}$$

Here, $\alpha_j$ is a "group-level" intercept capturing the quality of the base solution at $N_j$, while $\sigma_y \epsilon_{\tilde{N}}$ represents per-instance noise. The GEN node (action $a_0$) is treated as a newly introduced group without its own direct observations. However, its group-level intercept $\alpha_0$ is inferred not from the prior alone but rather from the posterior distribution over $\mu_\alpha$ and $\sigma_\alpha$, which is informed by the other observed data.

```
import pymc as pm

# Child indices use 0-based indexing; note the difference from index j in
    Equations (9-10)
child_indices = [0, 0, 0, 1, 2, 2]
rewards = [0.8, 0.8, 1.0, 0, 0.2, 0.3]
coords = {"child_idx": [0, 1, 2]}

with pm.Model(coords=coords) as model:
    ## Priors
    mu_alpha = pm.Normal("mu_alpha", mu=0.5, sigma=0.2)

    sigma_alpha = pm.HalfNormal("sigma_alpha", sigma=0.2)
    sigma_y = pm.HalfNormal("sigma_y", sigma=0.3)
    ## Priors END

    eps_j = pm.Normal("eps_j", mu=0, sigma=1, dims="child_idx")
    alpha = mu_alpha + eps_j * sigma_alpha

    r = pm.Normal("r", mu=alpha[child_indices], sigma=sigma_y, observed=
    rewards)
```

Listing 1: AB-MCTS-M fitting model example code

```
eps_j_gen = pm.Normal("eps_j_gen", mu=0, sigma=1)
alpha_gen = mu_alpha + eps_j_gen * sigma_alpha
r_gen = pm.Normal("r_gen", mu=alpha_gen, sigma=sigma_y)
```

Listing 2: AB-MCTS-M GEN node reward modeling

We place priors on $(\mu_\alpha, \sigma_\alpha, \sigma_y)$ and estimate them via MCMC (Markov Chain Monte Carlo), then perform Thompson Sampling. Because the GEN group ($j = 0$) has no direct observations, its posterior remains more uncertain and thus encourages exploration. To illustrate how to estimate posterior predictives, in Listing 1, we provide PyMC [41] code corresponding to the example tree shown in Figure 2. Here, we adopted the same priors as in our experimental setting, as noted in Appendix B.2:

$$\mu_\alpha \sim \mathcal{N}(0.5, 0.2^2), \quad \sigma_\alpha \sim \mathcal{N}_{\text{half}}(0.2^2), \quad \sigma_y \sim \mathcal{N}_{\text{half}}(0.3^2), \tag{11}$$

In this implementation, the variable `alpha` is node-specific (indexed by `child_idx`), yet shares parameters `mu_alpha`, representing the overall average answer quality determined by the inherent difficulty of the task, and `sigma_alpha`, representing the variability in answer quality arising from the LLM's response diversity. Differences in answer quality among nodes are captured by the variable `eps_j`.

To compute the probability distribution for the GEN node, we introduce a slightly modified predictive model by adding an additional variable representing the GEN node reward, as shown in Listing 2.

Since `eps_j_gen` has no associated observed data, `r_gen` typically exhibits higher variance compared to `r`. Intuitively, `r_gen` incorporates both the variance arising from the refinement process and the inherent variability in answer generation at node $N$. This increased variance encourages greater exploration during Thompson Sampling.

After model fitting, the posterior predictive distributions of `r` (existing child nodes) and `r_gen` (GEN node) are utilized for Thompson Sampling.

### A.4 AB-MCTS-A Details: Parameter Update Rules

#### A.4.1 AB-MCTS-A (Gaussian) Parameter Update Rules

As for the parameter update rules, for the Gaussian case, we use a normal-inverse-$\chi^2$ prior:

$$p(r \mid \{r_n\}_{n=1}^N) = \mathcal{N}(r \mid \hat{m}, \frac{\sigma^2}{\hat{\kappa}})\chi^{-2}(\sigma^2 \mid \hat{\nu}, \hat{\tau}^2), \tag{12}$$

$$\hat{m} = \frac{\breve{\kappa}\breve{m} + N\bar{r}}{\hat{\kappa}}, \tag{13}$$

$$\hat{\kappa} = \breve{\kappa} + N, \tag{14}$$

$$\hat{\nu} = \breve{\nu} + N, \tag{15}$$

$$\hat{\nu}\,\hat{\tau}^2 = \breve{\nu}\,\breve{\tau}^2 + \sum_{n=1}^N (r_n - \bar{r})^2 + \frac{N\,\breve{\kappa}}{\breve{\kappa} + N}\,(\hat{m} - \bar{r})^2, \tag{16}$$

$$\bar{r} = \frac{1}{N}\sum_{n=1}^N r_n, \tag{17}$$

where $r_n$ is the observed score.

#### A.4.2 AB-MCTS-A (Beta) Parameter Update Rules

Alternatively, if $r \in [0, 1]$, we can use a Beta distribution with the following parameter update rules after observing $\{r_n\}_{n=1}^N$:

$$p(r \mid \{r_n\}_{n=1}^N) = B(r \mid \hat{\alpha}, \hat{\beta}), \tag{18}$$

$$\hat{\alpha} = \breve{\alpha} + \sum_{n=1}^N r_n, \tag{19}$$

$$\hat{\beta} = \breve{\beta} + \sum_{n=1}^N (1 - r_n), \tag{20}$$

where $B(\cdot \mid \alpha, \beta)$ denotes the Beta distribution. We note that, usually this update rule is used in conjunction with the Bernoulli trial, but here we directly use Beta distribution to model the score distribution. In practice, this parameter update rule worked well according to our experimental results.

### A.5 Walk-through Examples

We walk through an iteration of AB-MCTS-M and AB-MCTS-A on the example trees in Figures 2 and 3. The process is stochastic due to Thompson sampling; for clarity, we assume specific sampled outcomes.

#### A.5.1 AB-MCTS-M

AB-MCTS-M incrementally builds the search tree by adding one node at a time. In this section, we detail a single iteration of the AB-MCTS-M algorithm, clearly illustrating the sequence of selecting a node to expand, performing the expansion, and backing up the resulting score. For simplicity and concreteness, we assume the current search tree structure is as depicted in Figure 2, and we describe one complete iteration, including selection, expansion, and score backup.

1. ($N \rightarrow N_1$) At $N$, we compute posterior distributions for its four children, GEN, $N_1$, $N_2$, and $N_3$ under the mixed model, and draw one score from each. If GEN receives the highest sample it is expanded and its score is backed up (Algorithm 1, lines 11–13). Here, we assume that $N_1$ attains the highest sampled score, reflecting the exploitation of a child whose posterior peak is comparatively large.

2. ($N_1 \rightarrow N_1'$) $N_1$ has two direct children, $N_1'$ ($r = 0.8$) and $N_2'$ ($r = 1.0$). We compute posteriors for GEN, $N_1'$, and $N_2'$, then sample again. Although the subtree $T(N_2')$ currently contains nodes with higher score as we can see from Figure 2, the finite variance of its posterior ensures that $N_2'$ is not always selected, and encourages more exploration for under-explored tree regions. Suppose $N_1'$ is chosen on this iteration.

3. **(Expanding $N_1'$)** Because $N_1'$ is a leaf, we expand it (Algorithm 1, line 10). Assume the newly generated node receives the score $r = 0.5$. Since all the leaf nodes have a GEN child, a GEN node is appended to this expanded node as well.

4. **(Score backup)** The score is propagated upwards from the expanded node toward the root, as in standard MCTS. In the current example, the score is backed up through the node generated at step 3, then through nodes $N_1'$, $N_1$, and $N$. Unlike standard MCTS, AB-MCTS-M maintains individual scores rather than averages. Specifically, the backed-up scores for nodes $N_1, N_2, N_3$ form distinct observation lists corresponding to each group in the mixed model. Although GEN nodes have no direct observations due to this score backup rule, their posterior distributions share statistical strength with these groups, allowing indirect information sharing and improved estimation accuracy for a score obtained by node expansion. At the next `SelectExpansionTarget` call, the four posteriors at $N$ have different shapes; specifically, the peak of $N_1$'s posterior shifts left due to the lowered expected value. Other posterior distributions are affected as well, e.g., the right-hand tail of the GEN posterior contracts. Please note that, in AB-MCTS-M, the change of posterior distribution shape cannot be analytically written down, and is calculated by MCMC. Thus, unlike standard MCTS, the score backup step involves appending the new score to a list of scores.

### A.5.2 AB-MCTS-A

AB-MCTS-A works in a similar manner to AB-MCTS-M, except for the introduction of CONT node and how we perform score backup. To clearly illustrate the algorithm and highlight the difference from AB-MCTS-M, we describe a complete iteration of selection, expansion, and score backup cycle for AB-MCTS-A. Since the only difference between Beta and Gaussian variants is how the score update is reflected in the posterior distribution, here we focus on the qualitative aspect of posterior update and focus on the details for an example tree, Figure 3.

1. **($N \rightarrow$ CONT)** At the root, we sample from the posteriors of the GEN and CONT children. As detailed in Section 3.4, the GEN posterior is informed by the CONT children nodes $N_1$ (0.8), $N_2$ (0.0), $N_3$ (0.2), whereas the CONT posterior uses the CONT node's descendant scores excluding $N_1$, $N_2$ and $N_3$, i.e., 0.8, 1.0, 0.3. We assume CONT is selected here.

2. **(CONT $\rightarrow N_1$)** Next we compute posteriors for CONT's children $N_1$, $N_2$, and $N_3$. Due to the score-backup rule, the posterior distributions are computed from previously expanded nodes; Concretely, the following scores are used for posterior distribution calculation: $N_1$: (0.8, 0.8, 1.0), $N_2$: (0.0), and $N_3$: (0.2, 0.3). Here, we assume $N_1$ obtains the highest sample via Thompson sampling.

3. **(Expanding $N_1$)** Again, we perform Thompson sampling between the GEN and CONT children of $N_1$. The GEN posterior uses (0.8, 1.0); the CONT posterior falls back to the prior because no generated descendants exist. Here we assume GEN is selected, leading to the expansion of a new node under $N_1$'s CONT child. We assume the score $r = 0.5$.

4. **(Score backup)** We backup the score $r = 0.5$ as prescribed in Section 3.4. First, the score is backed up to the GEN node which generates the node. Second, it propagates to: (i) GEN node's ancestor $N_1$, and (ii) the CONT node that is $N_1$'s parent. Because 0.5 is lower than the existing scores 0.8, 1.0, the posterior peak of $N_1$ shifts left, reducing the probability that $N_1$ will be chosen again from CONT. Similarly, adding 0.5 to the existing scores (0.8, 1.0, 0.3) lowers the peak of the CONT posterior at $N$, thus decreasing the probability that CONT will be selected at $N$ in later iterations.

### A.6 Hyperparameter Sensitivity of AB-MCTS

This section presents a sensitivity analysis of the hyperparameters used in AB-MCTS. As discussed in Appendix B.2, the prior parameters were designed to be non-informative to minimize bias. Since the posterior distributions become increasingly data-driven as the search progresses, we hypothesized that the influence of the initial priors would be limited. Here, we empirically verify this hypothesis through an extensive analysis. We evaluated the sensitivity of AB-MCTS variants to their prior hyperparameters on LiveCodeBench, using GPT-4o with a generation budget of $2^4$.

Table 3: **Hyperparameter Sensitivity of AB-MCTS.** Pass@1 results for each prior setting. All values are averaged over five runs.

| AB-MCTS-M | | | | | |
|---|---|---|---|---|---|
| $\breve{m}$ | 0.0 | 0.4 | 0.5 | 0.6 | 1.0 |
| Pass@1 | $38.4 \pm 1.6$ | $37.3 \pm 0.4$ | $36.8 \pm 1.5$ | $37.5 \pm 1.5$ | $37.7 \pm 1.3$ |
| $\breve{\alpha}$ | 0.01 | 0.1 | 0.2 | 0.3 | 1.0 |
| Pass@1 | $38.4 \pm 1.3$ | $37.7 \pm 0.9$ | $36.8 \pm 1.5$ | $38.2 \pm 2.3$ | $38.6 \pm 1.8$ |
| $\breve{\tau}$ | 0.01 | 0.1 | 0.2 | 0.3 | 1.0 |
| Pass@1 | $37.3 \pm 2.0$ | $38.2 \pm 1.1$ | $37.1 \pm 1.2$ | $36.8 \pm 1.5$ | $39.5 \pm 1.3$ |

| AB-MCTS-A (Gaussian) | | | | |
|---|---|---|---|---|
| $\breve{m}$ | 0.0 | 0.1 | 0.5 | 1.0 |
| Pass@1 | $38.0 \pm 1.6$ | $37.0 \pm 1.4$ | $37.3 \pm 1.5$ | $37.7 \pm 0.7$ |
| $\breve{\kappa}$ | 0.001 | 0.5 | 1.0 | 10.0 |
| Pass@1 | $38.4 \pm 1.3$ | $37.3 \pm 1.5$ | $38.0 \pm 1.6$ | $37.7 \pm 0.7$ |
| $\breve{\nu}$ | 0.001 | 0.5 | 1.0 | 10.0 |
| Pass@1 | $38.2 \pm 1.3$ | $37.7 \pm 1.3$ | $38.0 \pm 1.6$ | $38.9 \pm 1.0$ |

| $\breve{\tau}^2$ | 0.05 | 0.1 | 0.2 | 0.5 | 1.0 |
|---|---|---|---|---|---|
| Pass@1 | $37.5 \pm 0.9$ | $38.0 \pm 1.6$ | $37.7 \pm 1.3$ | $37.9 \pm 0.8$ | $37.7 \pm 0.7$ |

| AB-MCTS-A (Beta) | | | | | |
|---|---|---|---|---|---|
| $\breve{\alpha}$ | 0.1 | 0.4 | 0.5 | 0.6 | 1.0 |
| Pass@1 | $37.3 \pm 1.5$ | $37.0 \pm 1.5$ | $37.5 \pm 1.3$ | $37.5 \pm 1.3$ | $37.7 \pm 1.2$ |
| $\breve{\beta}$ | 0.1 | 0.4 | 0.5 | 0.6 | 1.0 |
| Pass@1 | $38.4 \pm 1.8$ | $38.4 \pm 1.1$ | $37.5 \pm 1.3$ | $37.9 \pm 0.5$ | $37.9 \pm 1.4$ |

Each configuration was run five times ($n = 5$) to compute mean and standard deviation of Pass@1. Tables 3 summarize the results for AB-MCTS-M, AB-MCTS-A (Gaussian), and AB-MCTS-A (Beta) under various prior settings. Across all tested ranges, the performance remains stable, indicating low sensitivity to the initial hyperparameter values. This confirms that AB-MCTS is robust to the initialization of prior hyperparameters, and its performance is primarily governed by the data-driven posterior updates during search.

# B  Additional Experimental Details

## B.1  Tasks and Datasets

We evaluated our approach on four benchmarks: CodeContest [1], LiveCodeBench [14], the Abstraction and Reasoning Corpus (ARC) [17], and MLE-Bench [16]. All of these benchmarks feature tasks that are often solved via code generation. Each experiment was run multiple times using non-zero temperature to account for stochasticity ($n = 5$ for LiveCodeBench, $n = 3$ for CodeContest and ARC-AGI). For MLE-Bench, experiments were run with $n = 1$ due to the significant computational cost.

**CodeContest and LiveCodeBench** are well-established competitive programming benchmarks, both providing public tests and hidden tests. We use the public tests to calculate each node's score and the hidden tests for final evaluation. A solution is counted as correct only if it passes all hidden test cases for a given problem, and the success rate is defined as the fraction of problems for which the chosen solution is fully correct. The prompt templates we used are based on those from previous work [14].

In LiveCodeBench, we only use problems released between August and November 2024, aligning with the previous work [19] to prevent data contamination.

**ARC-AGI** requires discovering a shared transformation rule from multiple input-output examples, then using it to predict the output for a test input. Generating code from sample grids is a frequently used approach for ARC [7, 13, 42]. We instruct the LLM to infer a transformation rule from the provided input/output examples and generate corresponding Python code. Each node's score is determined by the fraction of examples it correctly transforms. The node that achieves the highest score is then used to transform the test examples; if the output exactly matches the ground truth, that node's score is set to 1. We evaluate our method on the same set of 100 public evaluation problems and prompts used in prior work [13].

**MLE-Bench** comprises practical machine learning tasks derived from Kaggle competitions. In order to enable fair comparisons [16], we adopt three low-complexity challenges (*Nomad2018 Predicting Transparent Conductors*, *Spooky Author Identification*, and *Random Acts of Pizza*). Each competition's training data is randomly split into 80% for training and 20% for validation. The validation set is used to obtain the scores for each node. We select a node with the highest validation score at a given inference budget and then evaluate it on the hidden test set to get the final result. Following previous research [16], we use the AIDE scaffold for our experiments. Evaluating the generated machine learning models within these competitions is notably resource-intensive, requiring substantial GPU power even to process a single solution candidate. In our experiments, each solution candidate was executed on a single H100 GPU with a time limit of one hour. This computational demand is still considerably higher than for other benchmarks discussed in this work. Consequently, due to these significant costs, comprehensive experimentation across all methods and models on MLE-Bench was prohibitive. We therefore focused our evaluations using GPT-4o for these tasks specifically on AB-MCTS-M.

## B.2 AB-MCTS Parameters

Due to its Bayesian nature, the hyperparameters of AB-MCTS consist only of prior parameters. We used the same prior parameters for all the tasks without task-specific domain knowledge (except for the range of the score being approximately $[0, 1]$), thereby minimizing any potential bias. Consequently, our priors have the following shared properties:

- The vast majority of the probability mass (or all of it, in the case AB-MCTS-A (Beta)) is within $[0, 1]$.

- The average value of score is $0.5$, reflecting a neutral initial assumption regarding answer quality (where $0$ indicates the worst and $1$ the best)

- The probability mass does not concentrate excessively in any particular region, reflecting our unbiased prior.

We assign the following priors for AB-MCTS-M in Equations 9 and 10:

$$\mu_\alpha \sim \mathcal{N}(0.5, 0.2^2), \quad \sigma_\alpha \sim \mathcal{N}_{\text{half}}(0.2^2), \quad \sigma_y \sim \mathcal{N}_{\text{half}}(0.3^2), \tag{21}$$

where $\mathcal{N}_{\text{half}}$ is the half-normal distribution (please refer to Section 3.3 and Appendix A.3). For AB-MCTS-A (Gaussian), we set $\breve{m} = 0$, $\breve{\kappa} = 1$, $\breve{\nu} = 1$, and $\breve{\tau}^2 = 0.1$ in Equations 12 - 17, and for AB-MCTS-A (Beta), we set $\breve{\alpha} = 0.5$ and $\breve{\beta} = 0.5$ in Equations 18 - 20. As we noted earlier, we chose these parameters in a way that imposes as few assumptions as possible to minimize bias.

We expect the dependency on specific initial prior parameter values to be minimal, largely due to the substantial computational budgets in our evaluations and because posterior distributions become increasingly data-dominated as the search tree expands with more score observations (up to $2^7$ nodes in most of the experiments and $2^9$ in the extended ARC-AGI experiments). Because our node selection methods (AB-MCTS-M, with its mixed model, and AB-MCTS-A using conjugate priors) are fundamentally Bayesian, the priors primarily influence the early stages of the search. Moreover, the "borrowing strength" mechanism inherent in the mixed model stabilizes posterior estimates and facilitates convergence. As the search progresses and score observations accumulate, the initial prior influence naturally diminishes.

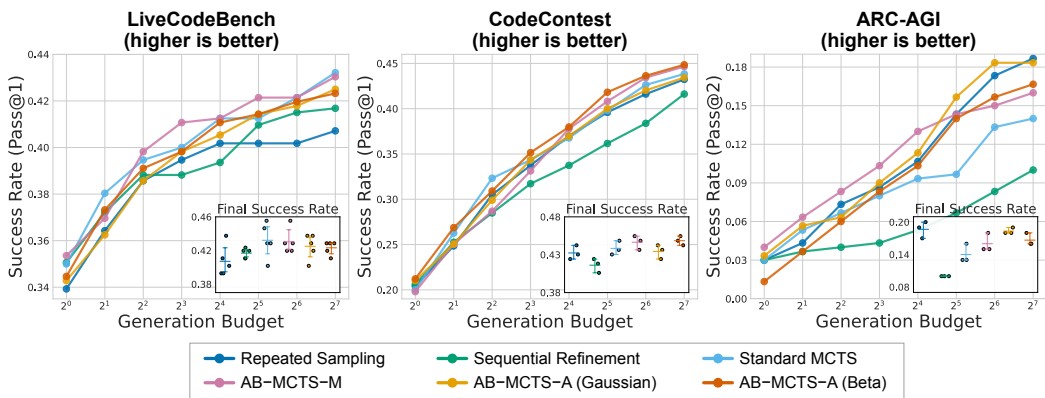

Figure 8: **Performance comparison with DeepSeek-V3 on LiveCodeBench, CodeContest, and ARC-AGI.** We compare AB-MCTS methods with the baselines by plotting the success rate against the generation budget.

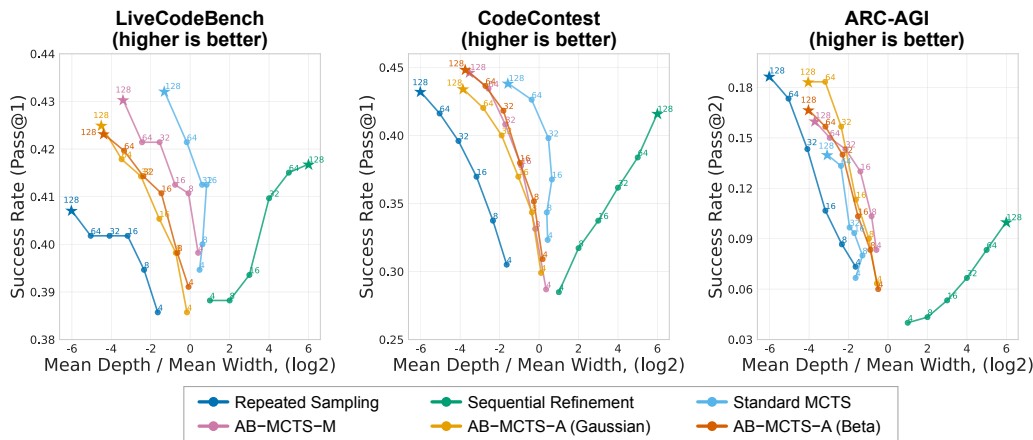

Figure 9: **Comparing algorithms by search tree shape and performance** Each point shows the performance against the average tree shape for a given algorithm at a specific generation budget. The x-axis represents the log-ratio of mean tree depth to mean tree width. Mean width is calculated as the average number of nodes per depth. Larger x-axis values indicate deeper searches, while smaller values indicate wider searches.

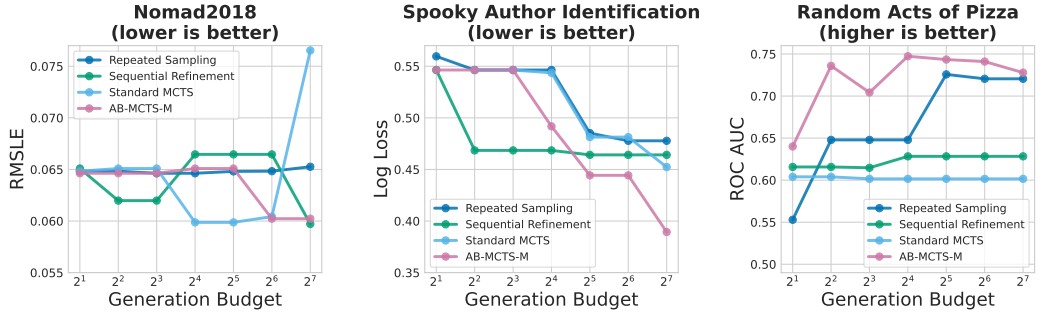

Figure 10: **Performance comparison on three MLE-Bench tasks using GPT-4o.** Each plot shows performance versus the total generation budget. For *Nomad2018 Predicting Transparent Conductors* and *Spooky Author Identification*, lower scores are better (RMSLE and Log Loss, respectively); for *Random Acts of Pizza*, higher is better (ROC AUC). At each budget, we choose the single solution based on validation-set performance and report its test-set score.

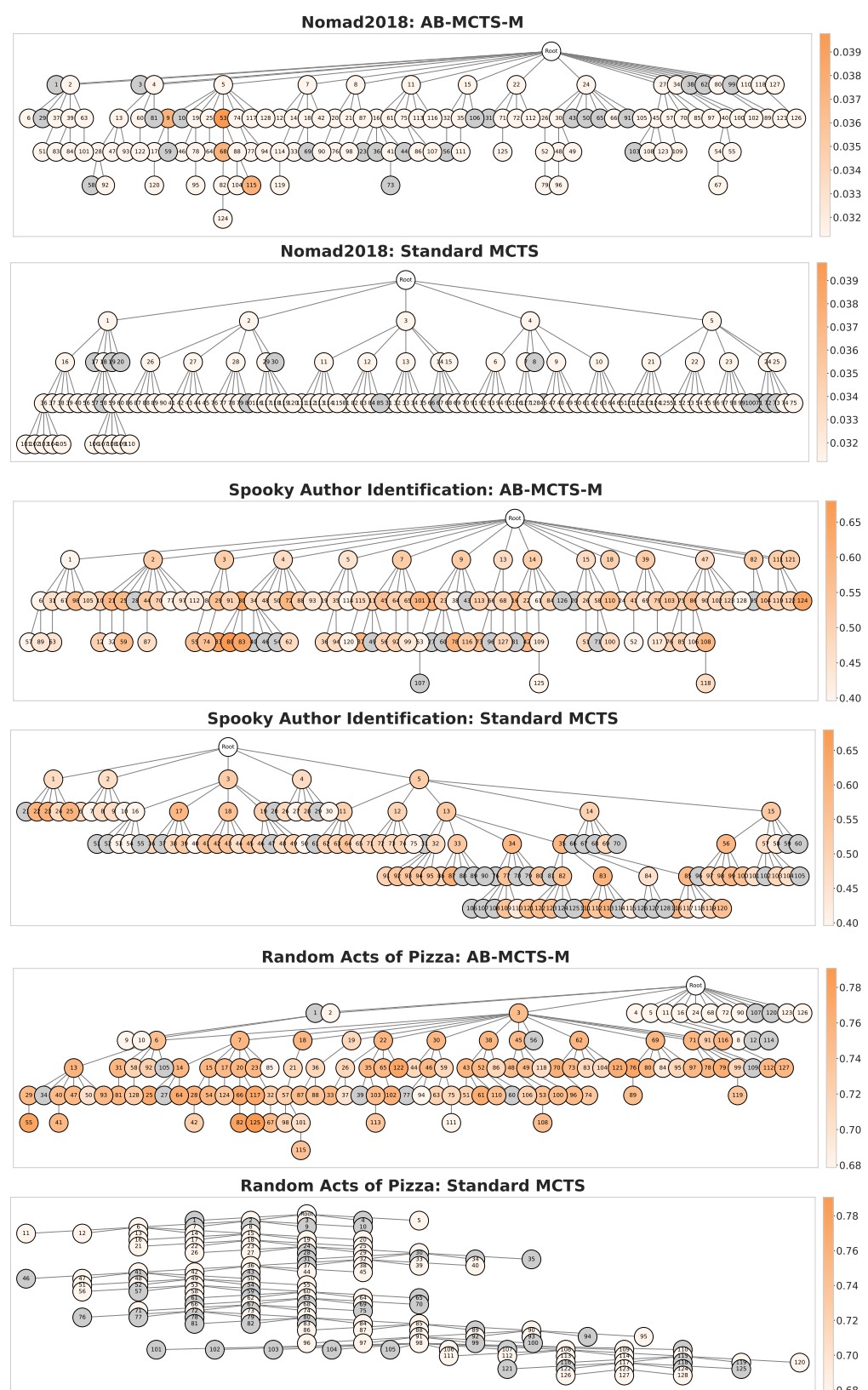

Figure 11: **Example search trees generated by AB-MCTS-M and standard MCTS on MLE-Bench.** The example tree for Random Acts of Pizza generated by AB-MCTS-M is the same tree shown in Figure 7

# C   Additional Experiments and Analysis

## C.1   Results with DeepSeek-V3 on Competitive Programming and ARC-AGI

This appendix provides supplementary results using the DeepSeek-V3 model, focusing on both overall performance and search behavior characteristics.

Figure 8 shows the performance (Pass@1) with DeepSeek-V3 on the Competitive Programming (LiveCodeBench, CodeContest) and ARC-AGI benchmarks. While the overall performance trends were similar to those with GPT-4o (Figure 4), the relative strengths of the baseline methods varied with DeepSeek-V3. For instance, standard MCTS achieved the highest success rate on LiveCodeBench. On CodeContest, the performance differences between AB-MCTS and the top-performing baselines were less pronounced than with GPT-4o. Despite these variations, AB-MCTS variants consistently placed among the leading methods across all tasks. This indicates that AB-MCTS reliably delivers strong performance even when changes in the underlying model alter the effectiveness of different baseline strategies.

The analysis of search tree shape versus performance with DeepSeek-V3 (Figure 9) closely mirrors the findings from GPT-4o (Figure 5). As expected, repeated sampling forms wide search trees, while sequential refinement develops deep trees. AB-MCTS methods consistently generated wider trees compared to standard MCTS. This tendency towards wider exploration is notable even on tasks like LiveCodeBench with DeepSeek-V3, where sequential refinement outperformed repeated sampling. The strong performance of AB-MCTS algorithms in such scenarios suggests their ability to not only explore broadly but also to effectively identify and deepen promising branches through their adaptive node selection. This highlights the robust nature of AB-MCTS in balancing these competing demands across different models.

## C.2   Results with GPT-4o on three competitions from MLE-Bench

Figure 10 compares AB-MCTS-M against the baseline methods on the three MLE-Bench competitions using GPT-4o, plotting performance scores against the generation budget. Notably, the most effective baseline varies across these competitions. For instance, on Nomad2018, sequential refinement ultimately achieves the best score while repeated sampling shows no improvement. On Spooky Author Identification, standard MCTS exhibits continued improvement throughout the budget range. Conversely, on Random Acts of Pizza, repeated sampling significantly outperforms other baselines. Despite this variability in baseline effectiveness, AB-MCTS-M consistently delivers strong performance across all three competitions. This highlights the robustness and adaptability of AB-MCTS-M, suggesting its capability to effectively adjust its search approach to the differing characteristics of each task, especially when the optimal strategy is not apparent beforehand.

## C.3   Example search trees generated by each methods on MLE-Bench

Figure 11 presents example search trees generated by AB-MCTS-M and standard MCTS for the three MLE-Bench competitions. Across all tasks, the trees generated by AB-MCTS-M visually suggest a more flexible approach compared to standard MCTS, effectively combining broader exploration with focused exploitation of promising nodes.

Table 4: **Comparison between Progressive Widening and AB-MCTS on LiveCodeBench.**

| | $(k, \alpha) = (1, 0.45)$ | $(k, \alpha) = (5, 0.5)$ | $(k, \alpha) = (10, 0.55)$ | AB-MCTS-A (Gaussian) | AB-MCTS-A (Beta) | AB-MCTS-M |
|---|---|---|---|---|---|---|
| Pass@1 | 48.7 ±0.8 | 50.7 ±1.3 | 50.5 ±3.1 | 48.9 ±2.0 | **51.8** ±1.4 | 49.6 ±1.6 |

## C.4   Comparison to Progressive Widening on LiveCodeBench

To compare the progressive widening with AB-MCTS, we conducted an experiment on Live-CodeBench for progressive widening with various parameters. We used `deepseek-v3-0324` with a generation budget $2^7$ ($n = 5$). The results are shown in Table 4. While an adequate progressive widening parameter leads to strong performance comparable to AB-MCTS, its effectiveness is highly sensitive to hyperparameters. For example, it produces the worst result with $(k, \alpha) = (1, 0.45)$.

Table 5: **Comparison of Pass@1 and Pass@2 on ARC-AGI with GPT-4o.**

| Method | Pass@1 | Pass@2 |
|---|---|---|
| Repeated Sampling | $14.0 \pm 1.7$ | $15.0 \pm 1.0$ |
| Sequential Refinement | $7.7 \pm 0.6$ | $8.7 \pm 0.9$ |
| Standard MCTS | $8.0 \pm 1.0$ | $9.0 \pm 1.5$ |
| AB-MCTS-M | $11.0 \pm 1.0$ | $12.3 \pm 1.2$ |
| AB-MCTS-A (Gaussian) | $13.0 \pm 3.6$ | $13.0 \pm 3.6$ |
| AB-MCTS-A (Beta) | $12.7 \pm 0.6$ | $14.0 \pm 2.1$ |

Furthermore, the $(k, \alpha) = (10, 0.55)$ setting shows search instability, leading to the highest variance. In contrast, AB-MCTS is robust without such tuning, demonstrating its practical advantage.

### C.5   AB-MCTS-M vs. AB-MCTS-A: Analysis and Selection

This paper introduces two adaptive branching algorithms: AB-MCTS-M and AB-MCTS-A. This section offers considerations for selecting between them, drawing upon our experimental findings and their distinct underlying mechanisms.

When outcome quality is the main priority, AB-MCTS-M is often the preferred choice because of its consistently strong performance (Table 1 and 2). For node selection, this algorithm uses MCMC, an iterative procedure that improves effectiveness but incurs some computational cost per selection. For applications with strict time constraints, AB-MCTS-A offers a lighter alternative, as its Gaussian and Beta variants employ analytically tractable posterior updates. However, when the LLM-inference time or the time required to evaluate the generated candidate solutions dominate, the extra time spent on MCMC has little impact on total runtime.

While both algorithms feature adaptive search, Figures 5 and 9 show that AB-MCTS-A tends to construct wider search trees. This tendency is attributed to its core design: at each depth, AB-MCTS-A chooses between selecting a GEN node or a CONT node. Reaching depth $d$ requires $d$ consecutive CONT choices, so deeper paths become geometrically less likely than wider expansions (see also Section A.4). Therefore, for tasks such as ARC-AGI where broader exploration is considered particularly beneficial, AB-MCTS-A can be preferable.

Ultimately, the choice depends on the application's primary goal: AB-MCTS-M often performs well when outcome quality is prioritized, whereas AB-MCTS-A offers advantages in computational efficiency and for tasks that benefit from broader exploration. However, both provide capable adaptive search strategies.

### C.6   Pass@1 vs. Pass@2 on ARC-AGI

In Table 1, we reported Pass@2 scores for ARC-AGI to follow the standard evaluation protocol defined by the benchmark [17]. For completeness and transparency, we also report the corresponding Pass@1 results in Table 5. As shown in Table 5, the results indicate consistent relative performance between Pass@1 and Pass@2. In particular, the ranking of methods remains unchanged, confirming that the conclusions presented in the main text are robust to the evaluation metric.

### C.7   Analysis of Adaptive Search Behavior via Node Degree Distribution

To gain a granular view of the adaptive nature of our methods, we analyze the node degree distribution of the search trees generated on LiveCodeBench with DeepSeek-V3, illustrated in Figure 12. The results show two distinct behaviors. AB-MCTS-M clearly favors a depth-focused search, with around 90% of its non-leaf nodes having a small degree (1–3). However, its long-tail distribution, with degrees up to 40, confirms that it adaptively broadens the search when required. In contrast, AB-MCTS-A performs a much wider search. Low-degree nodes (1–3) account for only about 30% of its non-leaf nodes, and the distribution spreads broadly, with degrees exceeding 100. This adaptive shaping of the search tree, which differs starkly from the more rigid patterns of baselines, provides strong evidence for the flexibility of our framework.

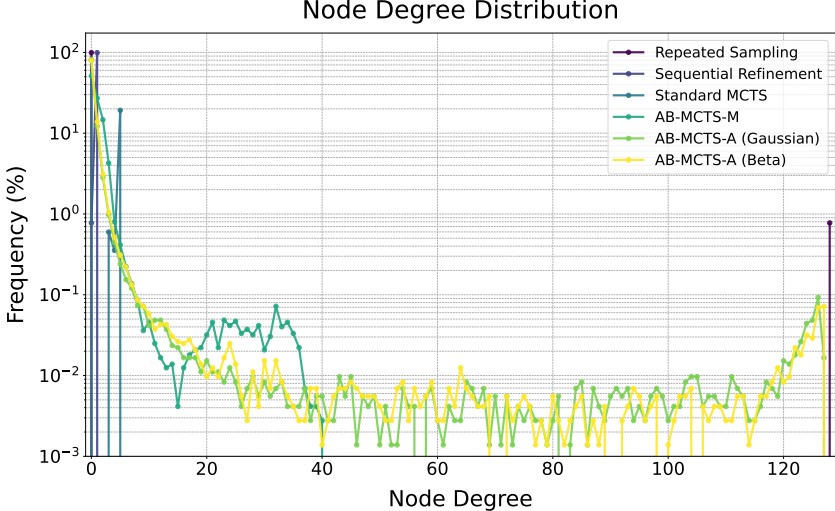

Figure 12: **Node-degree distribution reveals the adaptive search nature of our methods.** The plot displays the frequency of nodes (y-axis, log scale) for each degree (x-axis) under a search budget of $N = 128$.

Table 6: **Performance of Standard MCTS with Different Fixed Branching Factors ($w$) on LiveCodeBench.** Pass@1 scores are reported. The best result is shown in bold.

| Metric | $w = 3$ | $w = 5$ | $w = 10$ |
|---|---|---|---|
| Pass@1 | $0.429 \pm 0.018$ | $\mathbf{0.432 \pm 0.021}$ | $0.402 \pm 0.015$ |

## C.8 Ablation on the Fixed Branching Factor $w$ in Standard MCTS

We tested standard MCTS with fixed branching factors ($w$) of 3, 5, and 10 on LiveCodeBench using DeepSeek-V3. The results are summarized in Table 6. We find that $w = 5$ achieves the best performance among the tested values. Crucially, performance degrades with both smaller and larger widths, highlighting the sensitivity of the baseline to this hyperparameter. This finding underscores a key advantage of our adaptive method, which dynamically adjusts the effective branching factor during the search.

## C.9 Robustness of AB-MCTS Performance

In other baseline methods, the branching factor is either predetermined or determined by hyperparameters. For example, we need to predefine the branching factor in standard MCTS. However, the efficiency of the width vs depth strongly depends on the type of tasks and LLMs. This is reflected in Table 1 and Table 2, where AB-MCTS shows robust performance across various task types, while other methods excel for some tasks, but not for others. This is due to the adaptive branching nature of AB-MCTS, where the algorithm adapts to the wide or deep direction depending on the observed rewards. This is beneficial in the context of LLM inference-time scaling, since recently LLM has been used for solving various kinds of tasks, e.g., math tasks, coding tasks, etc., and a search algorithm that works for various task types out of the box is in high demand.

## C.10 Towards Pushing the Pareto Frontier of LLM Inference-Time Scaling

In this section, we propose the following two potential directions for future work aimed at pushing the Pareto frontier of LLM inference-time scaling:

1. **Enhanced Adaptivity via Difficulty Estimation:** We propose to enhance our algorithm's existing depth-width balancing by explicitly estimating problem difficulty from collected rewards. For difficult problems (identified by low rewards), the strategy would dynamically

switch, e.g., from a deep search (AB-MCTS-M) to a wide one (AB-MCTS-A). This is motivated by findings that optimal search strategies depend on difficulty [27].

2. **Collaborative Search with Multiple LLMs:** We also propose a method that leverages the diverse strengths of different LLMs within a single search. This is implemented by extending AB-MCTS with multiple GEN nodes, one for each LLM.

We elaborate on the specific methodology and preliminary experimental results for the second direction (Collaborative Search) in Section D.

# D   Multi-LLM AB-MCTS

Depending on the task, using more than one LLM for answer search [43] can be advantageous. For example, if LLM A can generate more diverse initial answers and LLM B excels at refinement, building the answer tree with both models is expected to improve performance. This section demonstrates how AB-MCTS can be extended to scenarios in which multiple LLMs are available for answer generation, and also reports the experimental results for ARC-AGI-2, demonstrating the effectiveness of the proposed method.

## D.1   Method

### D.1.1   Multiple LLMs as Answer Generators

Suppose $L$ LLMs are available for answer generation. We denote by $f_{\text{LLM}}^l$ the answer generator implemented by the $l$-th LLM, with $l = 1, \ldots, L$, following the notation introduced in Equation 2. The overall procedure is identical to the single-LLM AB-MCTS (Algorithm 1), with the addition of a step to select one of the $L$ available generators for node expansion. This selection occurs at each expansion phase, utilizing the current state of the answer tree $T$. The chosen generator, $f_{\text{LLM}}^l$, is then used to expand the selected node. We introduce two distinct algorithms for this generator selection process.

### D.1.2   Generator Selection Algorithm I: Single GEN Node

In this algorithm, node selection proceeds identically to AB-MCTS, followed by an additional generator selection step. During this step, each node $N$ in the entire answer tree $T$ is annotated with the index $l$ of the generator that produced it. For every generator, we obtain the set of nodes $\mathcal{N}_l \subseteq T$, containing all nodes it has generated; $\mathcal{N}_l$ is empty if the generator has not been selected yet. The scores of the nodes in $\mathcal{N}_l$ are used to calculate the posterior distribution of the expected score of a new node that will be generated by the generator $l$. After all $L$ posteriors have been computed, Thompson sampling is applied: a single sample is drawn from each distribution, and the generator with the highest sample is selected for the node expansion. These posteriors are modeled in the same way as the distributions used for node expansion target selection, as detailed below.

**Multi-LLM AB-MCTS-M with Generator Selection Algorithm I.** In this algorithm, Multi-LLM AB-MCTS-M treats the generators as groups within the mixed-effects model defined as

$$r_{\tilde{N}} = \alpha_l + \sigma_y \epsilon_{\tilde{N}}, \quad \alpha_l = \mu_\alpha + \sigma_\alpha \epsilon_l, \tag{22}$$
$$\epsilon_{\tilde{N}} \sim \mathcal{N}(0, 1), \quad \epsilon_l \sim \mathcal{N}(0, 1), \tag{23}$$

where the group index $j$ has been replaced by the generator index $l$ from Equations 9–10. As for the prior distributions, we can use the ones in Equation 11. After calculating the score posterior distributions using $\mathcal{N}_l$ for all $l$, we perform Thompson sampling to select one generator.

**Multi-LLM AB-MCTS-A with Generator Selection Algorithm I.** In this algorithm, Multi-LLM AB-MCTS-A assigns each $l$-th generator an independent prior–Gaussian (Equations 12-17) or Beta (Equations 18-20)–depending on the score metric, and updates these priors to posteriors using the scores of the nodes in $\mathcal{N}_l$. After calculating the posteriors for all $l$, we perform Thompson sampling to choose a generator.

### D.1.3 Generator Selection Algorithm II: Multiple GEN Nodes

An alternative approach attaches multiple GEN nodes as children to every node in the tree–one for each of the $L$ available generators. At each node $N$ in the expansion target selection process, the following process occurs:

1. The AB-MCTS selection logic is applied independently within the sub-trees associated with each generator. This means that for each generator $l$, we run a selection process considering its associated GEN node and any child nodes previously generated by it from node $N$.

2. Thompson sampling is used to identify the best node (either the GEN node or an existing child node) for each generator $l$.

3. Finally, the scores of the $L$ best nodes selected in the previous step are compared, and the one with the highest overall score is selected.

This algorithm is designed to better capture the local context of the search tree, allowing for the adaptive selection of the most suitable generator at each specific stage of the solution process.

## D.2 Experiments

In this section, we evaluate the effectiveness of Multi-LLM AB-MCTS by reporting the results and analysis of ARC-AGI-2 [44]. ARC-AGI-2 is an enhanced benchmark building upon the original ARC-AGI, specifically designed to assess higher-level cognitive abilities of artificial intelligence systems rigorously. It maintains the same fundamental principles as ARC-AGI, emphasizing tasks that require general fluid intelligence rather than extensive prior knowledge or memorization. We selected this benchmark, which is hard to solve even for frontier LLMs (less than 5% success rate [44]), to assess whether we can combine multiple frontier reasoning LLMs to obtain better performance for challenging tasks.

### D.2.1 Experimental Setup

In this experiment, we evaluated our approach on the 120 problems comprising the public evaluation set of ARC-AGI-2. For each problem, the generation budget was set to 250. The solution generation and refinement procedures are the same as those of ARC-AGI-1 experiment: The models were instructed to generate the transformation rule as Python code, and the search was guided by a reward signal corresponding to the number of demonstration cases correctly solved by the generated code. For solution generation, we used three frontier reasoning models: Gemini-2.5 Pro (`gemini-2.5-pro-preview-05-06`) [45], o4-mini (`o4-mini-2025-04-16`) [46], and DeepSeek-R1-0528 (`deepseek-r1-0528`) [22]. The temperature is set to be 0.6 for all the models.

To primarily evaluate the potential of Multi-LLM AB-MCTS, we used the Pass@k metric. This metric measures whether at least one correct solution is found within k attempts. This differs from the official ARC-AGI-2 contest standard, which typically uses a Pass@2 criterion (i.e., one of two submitted final answers must be correct). Evaluating Pass@2 requires an additional selection mechanism to identify promising candidates from the search history. Therefore, this experiment focuses on the search capability itself via Pass@k. Due to the significant API costs of the used frontier reasoning models, we focused on the evaluation of Single-LLM AB-MCTS-A and Multi-LLM AB-MCTS-A (and repeated sampling for o4-mini), since it worked most efficiently in our experiment on ARC-AGI-1. Also, we employed the generator selection algorithm II (see Section D.1.3 for details) for our experiments.

### D.2.2 Results

The performance of our proposed methods was compared against a repeated sampling, which was the most efficient method for the ARC-AGI-1 experiment, and the result is shown in Fig. 13. As shown, Repeated Sampling using the o4-mini model achieved a 23% Pass@k success rate on the public evaluation set, and the single-model AB-MCTS using o4-mini improved the success rate to 27.5%. The performance advantage of AB-MCTS over Repeated Sampling becomes more evident as the generation budget increases, particularly after approximately 50 budget.

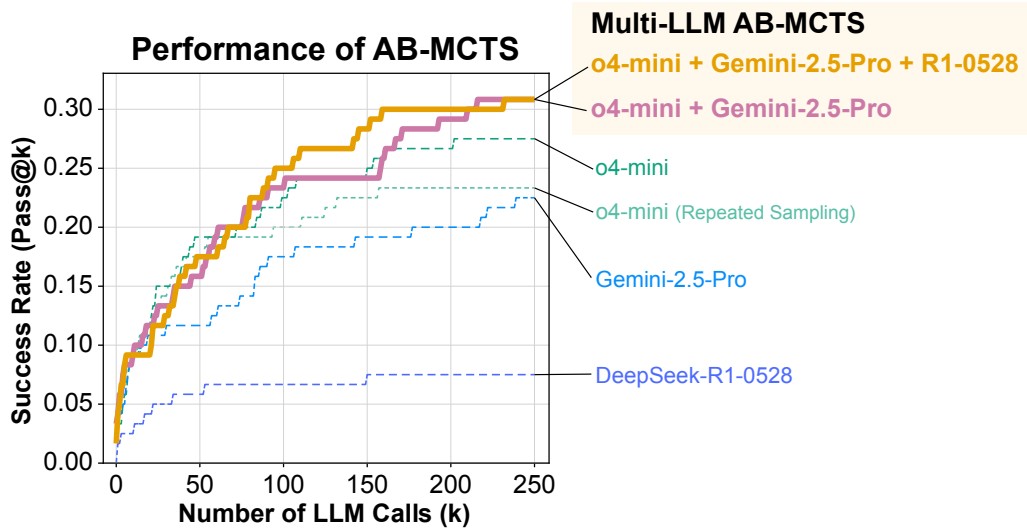

Figure 13: Pass@k (coverage) of ARC-AGI-2 for each generation budget. The methods tested are repeated sampling, AB-MCTS, and Multi-LLM AB-MCTS methods.

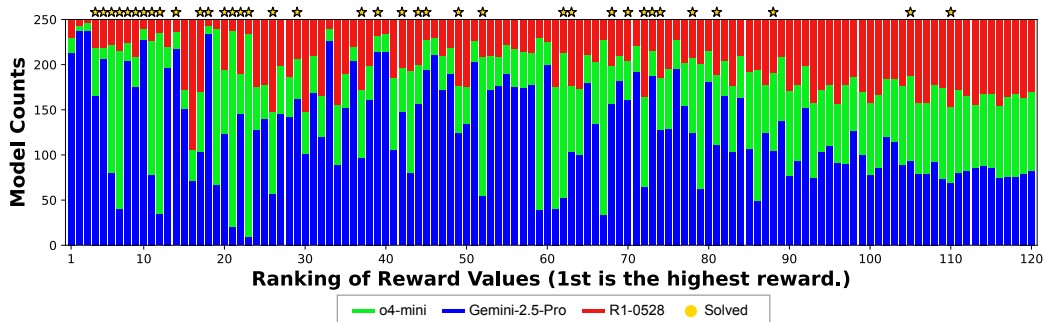

Figure 14: Distribution of LLM usage in the 120 ARC-AGI-2 problems with Multi-LLM AB-MCTS-A. Trials are sorted by the maximum reward obtained from demonstration cases (higher reward to the left). Starred trials indicate a correct final solution was found.

By employing Multi-LLM AB-MCTS, which integrates Gemini-2.5-Pro and DeepSeek-R1-0528, we further improved the performance, ultimately finding correct solutions for over 30% of the problems. Notably, although DeepSeek-R1-0528 exhibited lower individual performance, its integration into the Multi-LLM framework led to an increase in the number of solved problems.

### D.2.3 Analysis

The Multi-LLM AB-MCTS framework demonstrated an ability to effectively allocate different LLMs to problems based on their characteristics. As illustrated in Figure 14, the distribution of LLM usage varied across problems. For trials that quickly achieved a high reward from the demonstration cases (left side of the figure), the more proficient model tended to be assigned. Conversely, for trials where obtaining a high reward was more challenging (right side), the models were utilized in a more balanced manner.

Furthermore, we observed instances where problems unsolvable by any single LLM were solved through the collaboration of multiple models. This suggests a synergistic interaction that transcends simply matching the best model to a problem. Figure 15 and Figure 16 depict a search process where an incorrect solution generated by o4-mini served as a useful hint for DeepSeek-R1-0528 and Gemini-2.5-Pro, which then collaboratively produced the correct solution. This result indicates

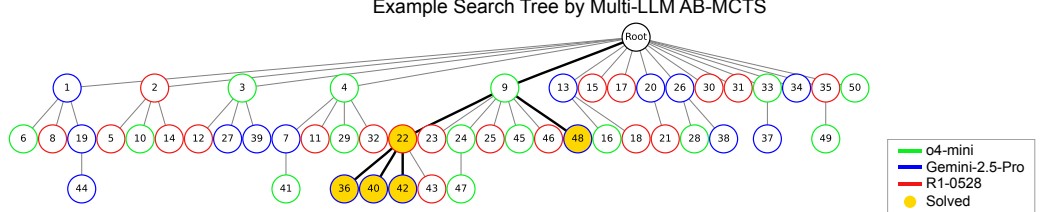

Figure 15: An example search tree from a successful trial on ARC-AGI-2 using Multi-LLM AB-MCTS-A. The number in each node indicates the generation step, and the color represents the selected LLM. The yellow node generated the code that correctly solved the test case. This problem was not solved by any single model in isolation.

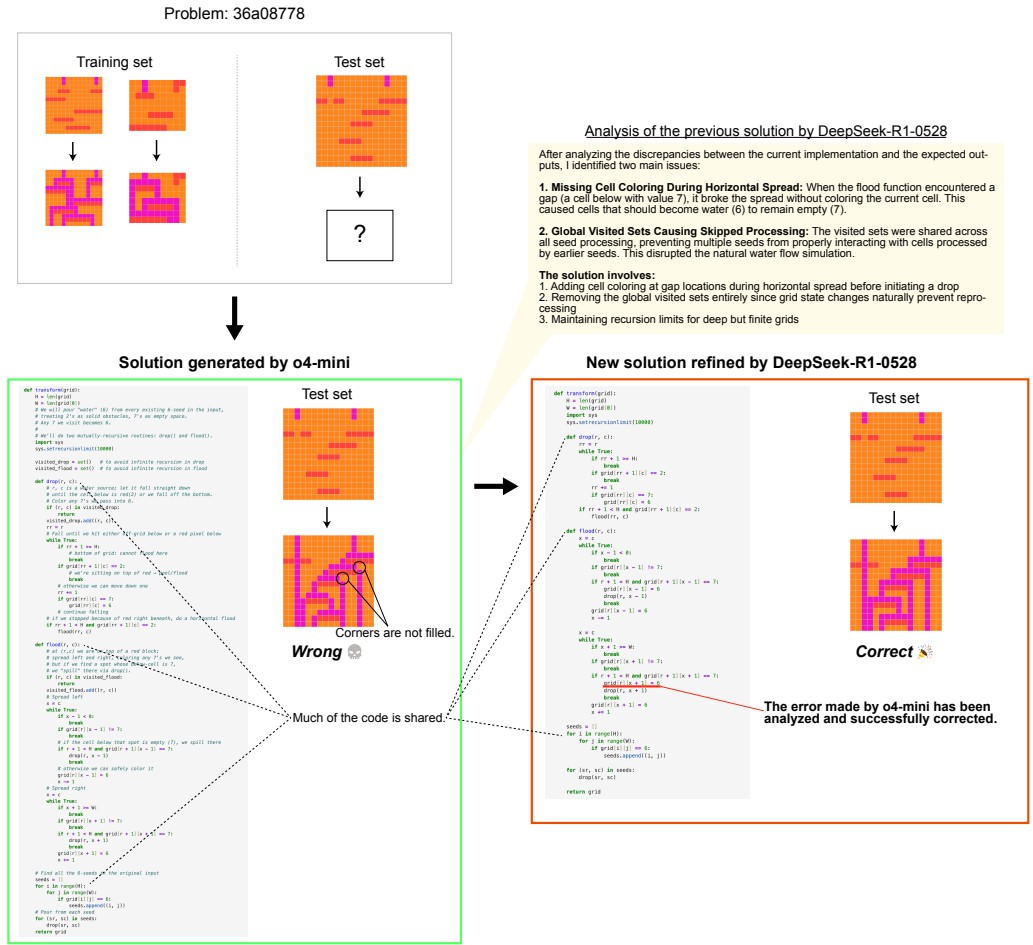

Figure 16: An illustration of model collaboration. In this example, DeepSeek-R1-0528 refines an incorrect intermediate solution generated by o4-mini (from the problem shown in Figure 15) to produce the final correct solution.

that Multi-LLM AB-MCTS can facilitate flexible and effective collaboration among heterogeneous frontier LLMs.

### D.3 Challenges and Future Work

While our primary evaluation focused on search capability using the Pass@k metric, we conducted a preliminary evaluation based on the Pass@2 criterion for reference. Using a simple rule-based method to select two final answers (prioritizing code with a high reward generated later in the search), the Multi-LLM AB-MCTS achieved a Pass@2 of 19.2%. Although this is a promising result, a significant gap of over 10 percentage points remains compared to the 30% Pass@250 rate. Future work should focus on closing this gap by developing more sophisticated final-answer selection algorithms. Potential directions include building more accurate reward models or integrating an LLM-as-a-Judge for a more nuanced evaluation of candidate solutions.

