# OpenReview forum: "Wider or Deeper?  Scaling LLM Inference-Time Compute with Adaptive Branching Tree Search"
_NeurIPS.cc/2025/Conference — NeurIPS 2025 spotlight_

### Official Review · Reviewer_hmeP · 2025-06-30

**Clarity:** 2
**Significance:** 3
**Originality:** 4
**Rating:** 5
**Confidence:** 3

**Summary:**

The paper proposes  modified forms of MCTS. In standard MCTS, only leaf nodes are expanded and a fixed set of child nodes are expanded. The modified versions of MCTS instead add a child node (to every node) called GEN, which when called, expands the parent node again. Two selection policies are introduced called AB-MCTS-M and AB-MCTS-A that use either a mixed model for scoring or aggregate node values and are estimated with Gaussian or Beta distributions. The different versions are evaluated on different tasks and perform better than different baselines.

**Questions:**

- What is the computation cost from adaptive MCTS?
- Why are there two versions of adaptive MCTS? Is one better for some applications than others?

**Ethical Concerns:**

["NO or VERY MINOR ethics concerns only"]

**Final Justification:**

My main concerns were the small increase in performance and the computational overhead. The authors addressed this with statistical significance tests in another rebutal response and provided computational costs for the different methods on a CPU.

The paper identifies and generates a unique solution to an existing problem that has not been addressed. Given the popularity of MCTS techniques, it is an important development and would be of great interest to the community.

**Limitations:**

yes

**Quality:**

3

**Strengths And Weaknesses:**

Strengths:
- The new MCTS variants introduced are novel and a creative solution to fixed-width MCTS. (Originality)
- The paper identifies an important weakness of MCTS which is that it favors exploration over exploitation. (Significance)
- The methods do well under smaller generation budget. (Quality)

Weaknesses:
- While the conceptual motivation is clear, the experimental results do not show large gains from the approach making it hard to justify the extra complexity. (Quality)
- It is not clear what the computation cost is from the new methods compared to typical MCTS. (Clarity)

---

> ### Author Rebuttal · Authors · 2025-07-31
>
> We would like to thank the reviewer for the thoughtful feedback on our manuscript. We have carefully addressed **all the weaknesses and questions** raised. Although the rebuttal format precludes the submission of a revised manuscript at this stage, we have incorporated all the necessary changes into the manuscript based on the feedback. The key revisions we have made are summarized below:
>
> * **[W1] Performance Gains:** The manuscript has been revised to further emphasize that AB-MCTS exhibits superior robustness across a diverse range of tasks.
> * **[W2/Q1] Computational Overhead:** We measured the overhead and found that it is not the practical bottleneck.
> * **[Q2] Rationale for Two AB-MCTS Designs:** We have clarified the rationale for our two designs by elaborating on the different trade-offs for each method.
>
> We elaborate on each of these points below.
>
> ---
>
> ## [W1] Performance Gains
>
> **TLDR:** AB-MCTS exhibits superior robustness across a diverse range of tasks.
>
> > While the conceptual motivation is clear, the experimental results do not show large gains from the approach making it hard to justify the extra complexity. (Quality)
>
> We thank the reviewer for the opportunity to clarify the advantage of AB-MCTS. As shown in the `Avg. Rank` columns in Tables 1 and 2, the strength of our method lies in its robustness, delivering top-level performance across a variety of tasks. We revised our manuscript to further clarify this point in Appendix A.2 as follows:
>
> In other baseline methods, the branching factor is either predetermined or determined by hyperparameters. For example, we need to predefine the branching factor in standard MCTS. However, the efficiency of the width vs depth strongly depends on the type of tasks and LLMs. This is reflected in Table 1 and Table 2, where AB-MCTS shows robust performance across various task types, while other methods excel for some tasks, but not for others. This is due to the adaptive branching nature of AB-MCTS, where the algorithm adapts to the wide or deep direction depending on the observed rewards. This is beneficial in the context of LLM test-time scaling, since recently LLM has been used for solving various kinds of tasks, e.g., math tasks, coding tasks, etc., and a search algorithm that works for various task types out of the box is in high demand.
>
> ---
>
> ## [W2/Q1] Computational Overhead
>
> **TLDR:** We measured the overhead and found that it is not the practical bottleneck.
>
> > It is not clear what the computation cost is from the new methods compared to typical MCTS. (Clarity)
>
> > What is the computation cost from adaptive MCTS?
>
> We thank the reviewer for the insightful comment, which is crucial for clarifying the properties of AB-MCTS. To quantify the computational overhead, we measured the execution times of our proposed methods against Standard MCTS.
>
> We used the same machine used in our experiments. Since all algorithms are CPU-intensive, we measured the wall-clock time. The experiments were conducted with a generation budget of 128, and for Standard MCTS, we used a branching factor of 5. Reward values are drawn from a uniform random distribution $U[0,1)$.
> The results from 50 runs are presented below, showing the average execution time per one MCTS step. All units are in milliseconds (ms).
>
> |Algorithm|Mean|Stdev|Minimum|Maximum|
> |:-|-:|-:|-:|-:|
> |Standard MCTS|$1.28 \times 10^{-2}$|$9.54 \times 10^{-4}$|$1.21 \times 10^{-2}$|$1.73 \times 10^{-2}$|
> |AB-MCTS-M|$1.56 \times 10^{4}$|$4.47 \times 10^{2}$|$1.48 \times 10^{4}$|$1.68 \times 10^{4}$|
> |AB-MCTS-A (Gaussian)|$2.28 \times 10^{0}$|$5.37 \times 10^{-1}$|$1.14 \times 10^{0}$|$3.3 \times 10^{0}$|
> |AB-MCTS-A (Beta)|$1.91 \times 10^{0}$|$3.27 \times 10^{-1}$|$1.23 \times 10^{0}$|$2.62 \times 10^{0}$|
>
> The execution time for AB-MCTS-A, regardless of the prior used, is approximately 2 ms, which is about x150 longer than Standard MCTS (0.013 ms). On the other hand, AB-MCTS-M shows a substantial increase compared to Standard MCTS and AB-MCTS-A, averaging 15600 ms.
> However, this overhead must be evaluated in the context of the entire system, particularly one that involves LLM calls. A single call to an LLM typically takes from tens of seconds to several minutes.
> Assuming a typical value of 1 minute per call, the overhead of AB-MCTS-A is negligible compared to the LLM call latency. Furthermore, even for the most computationally intensive method, AB-MCTS-M, its execution time (16 seconds) accounts for only about 26% of a LLM call time (60 seconds). Therefore, this suggests that the search process by AB-MCTS does not become the computational bottleneck of the overall process.
>
> While the execution time of AB-MCTS-M appears slow, this is primarily because its implementation was not actively optimized, as the practical bottleneck of the overall process lies in the time-consuming LLM calls and task evaluations. The bottleneck within AB-MCTS-M is its MCMC fitting process, which can be significantly accelerated, for example, through computation on a GPU.
>
> ---
>
> ## [Q2] Rationale for Two AB-MCTS Designs
>
> **TLDR:** We revised our paper to add elaboration on the different trade-offs for each method.
>
> > Why are there two versions of adaptive MCTS? Is one better for some applications than others?
>
> Thank you for the important and pragmatic question. We revised our manuscript to include a clarification on this point as a new section in the Appendix. We summarize the revision below.
>
> When outcome quality is the main priority, AB-MCTS-M is often the preferred choice because of its consistently strong performance (Tables 1 and 2). For node selection, this algorithm uses MCMC, an iterative procedure that improves effectiveness but incurs some computational cost per selection. For applications with strict time constraints, AB-MCTS-A offers a lighter alternative, as its posterior updates are done analytically. However, as the LLM-inference time or the time required to evaluate the generated candidate solutions often dominate, the extra time spent on MCMC has little impact on total runtime.
>
> While both algorithms feature adaptive search, Figures 5 and 9 show that AB-MCTS-A tends to construct wider search trees due to its core design: Reaching depth $d$ requires $d$ consecutive CONT choices, so deeper paths become geometrically less likely than wider expansions. Therefore, for tasks such as ARC-AGI, where broader exploration is considered particularly beneficial, AB-MCTS-A can be preferable.
>
> Ultimately, the choice depends on the application's primary goal: AB-MCTS-M often performs well when outcome quality is prioritized, whereas AB-MCTS-A offers advantages in computational efficiency and for tasks that benefit from broader exploration. However, both provide capable adaptive search strategies.
>
> ---
>
> Once again, we would like to thank the reviewer for the valuable feedback. We believe that our responses and the corresponding revisions have fully addressed all the concerns raised, and we hope the manuscript is now suitable for publication. Please do not hesitate to let us know if any further clarification is required.

---

> > ### Comment · Reviewer_hmeP · 2025-08-04
> >
> > Thank you to the authors for the comments and measurements. Based on the comments for Reviewer RYt1, I think the results do represent meaningful changes.  While there is the increase in computation, the increase can be mitigated with the appropriate hardware. Therefore, I change my score to 5.

---

> > > ### Author Response · Authors · 2025-08-05
> > >
> > > We sincerely thank the reviewer for their insightful feedback. Your comments significantly improved our manuscript’s clarity and practical relevance. We greatly appreciate your careful review and valuable suggestions.

---

### Official Review · Reviewer_Mkvu · 2025-07-01

**Clarity:** 3
**Significance:** 3
**Originality:** 3
**Rating:** 4
**Confidence:** 4

**Summary:**

This paper presents an adaptive-branching version of MCTS-based LLM inference to find the highest performing sequence under a fixed budget of LLM calls, given an external verifier. The approach uses a Bayesian selection strategy to either generate a new child node or refine an existing child at a particular node in the search tree based on an estimated posterior predictive and Thompson sampling.

**Questions:**

1. How are the parameters decided for the priors used in the mixed and aggregate models? Do you have experiments evaluating how sensitive the performance is to this choice?
2. Why is the width for standard MCTS fixed to 5, simply because previous works did that? For a fair comparison, it would be helpful to find the width (even a single setting across tasks) that performs the best and then show a comparison with the adaptive method.
3. Why are the standard deviations not reported despite repeat runs being executed? It would be useful to conclude whether the reported performance differences are significant or not.
4. While a reasonable choice, it is unclear to me how much the Bayesian selection mechanism adds value beyond a simpler UCT-style counting mechanism. Related, it remains unclear from the current description in the main text why progressive widening is not applicable in this setting. Could you provide an extended explanation?

**Ethical Concerns:**

["NO or VERY MINOR ethics concerns only"]

**Final Justification:**

The authors have provided additional experiments and analyses, which do indicate merit in the method. Additionally, the extended explanations vis-a-vis prior work is convincing to the extent of meriting this new method. My score would have been higher if the empirical results were stronger. Overall, I think this could be a good contribution.

**Limitations:**

Yes.

**Paper Formatting Concerns:**

None.

**Quality:**

3

**Strengths And Weaknesses:**

**Strengths:** I like the motivation of the paper and think the presented approach makes sense given that each problem instance may require very different search tree shapes to behave optimally. The paper is written clearly, and the experiments seem thorough.

**Weaknesses:** It would've been useful to tune the standard MCTS baseline to provide a more robust comparison, which would have also strengthened the case for wanting a dynamic branching factor. Additionally, standard deviations for the experiment results would have been very useful since temperature-based LLM sampling can have high variance.

---

> ### Author Rebuttal · Authors · 2025-07-31
>
> We are grateful for the reviewer's valuable feedback on our manuscript. We have carefully addressed **all the weaknesses and questions** raised. Although we cannot submit a revised version at this stage due to the rebuttal format, we have already implemented all the necessary revisions based on the reviewer's suggestions. Below is a summary of our responses and new results:
>
> * **[W1/Q2] Tuning the MCTS Baseline:** Provided new experiments to vary the MCTS baseline's branching factor.
> * **[W2/Q3] Reporting on Result Variance:** Added standard deviations to result tables, confirming their statistical reliability.
> * **[Q1] Hyperparameter Sensitivity of AB-MCTS:** Conducted experiments to analyze hyperparameter sensitivity and confirmed the method's robustness.
> * **[Q4-1] Rationale for a Non-UCT Approach:** UCT score cannot be used because of unbounded branching, while the Bayesian method can incorporate it naturally.
> * **[Q4-2] Comparison with Progressive Widening:** Compared AB-MCTS with progressive widening in new experiments, showing our method is more robust and consistent.
>
> We elaborate on each of these points below.
>
>
> ---
>
> ## [W1/Q2] Tuning the MCTS Baseline
>
> **TLDR:** Provided new experiments to vary the MCTS baseline's branching factor.
>
> Following the reviewer's valuable suggestion, we conducted experiments to vary the MCTS baseline's branching factor. Our initial choice of branching factor $w=5$ was based on its common use as a standard baseline setting in prior work. However, we agree that running this additional experiment provides a fairer comparison and highlights the benefits of our adaptive method.
>
> We tested standard MCTS with fixed branching factors ($w$) of 3, 5, and 10 on LiveCodeBench using `deepseek-v3`. The results are as follows:
> ||$w=3$|$w=5$|$w=10$|
> |-|-|-|-|
> |Pass@1|0.429±0.018|0.432±0.021|0.402±0.015|
>
> The results confirm that $w=5$ (the setting from our original manuscript) is indeed the optimal fixed width among those tested. Crucially, performance degrades with other settings, highlighting the sensitivity of the baseline to this hyperparameter. This finding underscores a key advantage of our adaptive method.
>
> We have revised our manuscript to add these experimental results to Appendix C. We are grateful for this suggestion, as it has significantly strengthened our experimental evaluation.
>
> ---
>
> ## [W2/Q3] Reporting on Result Variance
>
> **TLDR:** Added standard deviations to result tables, confirming their statistical reliability.
>
> We agree with the reviewer that including standard deviations is crucial for providing a clearer and more reliable picture of our experimental outcomes.
> We have updated our main results table (Table 1 in the paper) to include standard deviations, as shown below:
> |Method|LiveCodeBench (GPT-4o)|LiveCodeBench (DeepSeek-V3)|CodeContest (GPT-4o)|CodeContest (DeepSeek-V3)|ARC-AGI (GPT-4o)|ARC-AGI (DeepSeek-V3)|Avg. Rank|
> |-|-|-|-|-|-|-|-|
> |Repeated Sampling|37.8±0.5 (4)|40.7±1.9 (6)|37.9±0.3 (4)|43.2±0.9 (5)|15.0±1.0 (1)|18.6±1.0 (1)|3.5
> |Sequential Refinement|37.8±2.4 (4)|41.6±0.6 (5)|30.1±0.3 (6)|41.6±0.9 (6)|8.7±0.9 (6)|10.0±0.6 (6)|5.5
> |Standard MCTS|36.7±1.0 (6)|43.2±2.1 (1)|37.5±0.0 (5)|43.8±0.9 (3)|9.0±1.5 (5)|14.0±1.5 (5)|4.2
> |AB-MCTS-M|38.9±1.9 (2)|43.0±1.5 (2)|40.6±1.0 (1)|44.6±0.9 (2)|12.3±1.2 (4)|16.0±1.0 (3)|2.3
> |AB-MCTS-A(Gaussian)|39.1±1.9 (1)|42.5±1.5 (3)|40.2±1.7 (3)|43.4±0.9 (4)|13.0±3.6 (3)|18.3±0.6 (2)|2.7
> |AB-MCTS-A(Beta)|38.7±1.2 (3)|42.3±0.8 (4)|40.4±0.3 (2)|44.8±0.6 (1)|14.0±2.1 (2)|16.6±0.6 (4)|2.7
>
> The updated table confirms that the strong performance of our AB-MCTS methods is consistent across multiple runs. This addition makes our experimental findings significantly more robust.
>
> ---
>
> ## [Q1] Hyperparameter Sensitivity of AB-MCTS
>
> **TLDR:** Conducted experiments to analyze hyperparameter sensitivity and confirmed the method's robustness.
>
> The reviewer asks two important questions regarding our priors: (1) how these hyperparameters were chosen and (2) how sensitive performance is to this choice. We address both points below.
>
> As we state in Appendix B.2, we chose the prior parameters to be as non-informative as possible to minimize bias. We hypothesized that the influence of these initial priors would be minimal, as the posterior distributions become increasingly dominated by observed data as the search progresses (see Appendix B.2). However, we agree with the reviewer on the importance of empirically verifying this.
>
> To investigate this, we conducted a comprehensive sensitivity analysis. We ran new experiments on LiveCodeBench with a maximum generation budget of 16, using `gpt-4o` ($n=5$ runs), while varying the hyperparameters for each AB-MCTS variant.
>
> The results are presented below:
>
> **Table 1. AB-MCTS-M, $\breve{m}$**
> |$\breve{m}$|0.0|0.4|0.5|0.6|1.0|
> |-|-|-|-|-|-|
> |Pass@1|38.4±1.6|37.3±0.4|36.8±1.5|37.5±1.5|37.7±1.3|
>
>
> **Table 2. AB-MCTS-M, $\breve{\alpha}$**
> |$\breve{\alpha}$|0.01|0.1|0.2|0.3|1.0|
> |-|-|-|-|-|-|
> |Pass@1|38.4±1.3|37.7±0.9|36.8±1.5|38.2±2.3|38.6±1.8|
>
> **Table 3. AB-MCTS-M, $\breve{\tau}$**
> |$\breve{\tau}$|0.01|0.1|0.2|0.3|1.0|
> |-|-|-|-|-|-|
> |Pass@1|37.3±2.0|38.2±1.1|37.1±1.2|36.8±1.5|39.5±1.3|
>
> **Table 4. AB-MCTS-A (Gaussian), $\breve{m}$**
> |$\breve{m}$|0.0|0.1|0.5|1.0|
> |-|-|-|-|-|
> |Pass@1|38.0±1.6|37.0±1.4|37.3±1.5|37.7±0.7|
>
> **Table 5. AB-MCTS-A (Gaussian), $\breve{\kappa}$**
> |$\breve{\kappa}$|0.001|0.5|1.0|10.0|
> |-|-|-|-|-|
> |Pass@1|38.4±1.3|37.3±1.5|38.0±1.6|37.7±0.7|
>
> **Table 6. AB-MCTS-A (Gaussian), $\breve{\nu}$**
> |$\breve{\nu}$|0.001|0.5|1.0|10.0|
> |-|-|-|-|-|
> |Pass@1|38.2±1.3|37.7±1.3|38.0±1.6|38.9±1.0|
>
> **Table 7. AB-MCTS-A (Gaussian), $\breve{\tau}^2$**
> |$\breve{\tau}^2$|0.05|0.1|0.2|0.5|1.0|
> |-|-|-|-|-|-|
> |Pass@1|37.5±0.9|38.0±1.6|37.7±1.3|37.9±0.8|37.7±0.7|
>
>
> **Table 8. AB-MCTS-A (Beta), $\breve{\alpha}$**
> |$\breve{\alpha}$|0.1|0.4|0.5|0.6|1.0|
> |-|-|-|-|-|-|
> |Pass@1|37.3±1.5|37.0±1.5|37.5±1.3|37.5±1.3|37.7±1.2|
>
> **Table 9. AB-MCTS-A (Beta), $\breve{\beta}$**
> |$\breve{\beta}$|0.1|0.4|0.5|0.6|1.0|
> |-|-|-|-|-|-|
> |Pass@1|38.4±1.8|38.4±1.1|37.5±1.3|37.9±0.5|37.9±1.4|
>
> As shown in Tables 1-9 above, the performance is stable across a wide range of hyperparameter values, with most results falling within the standard deviation of each other. This confirms that our method is not sensitive to the initial choice of priors. We have revised the paper to include this full analysis in Appendix B.2 to more concretely demonstrate the robustness of our method.
>
> ---
>
> ## [Q4-1] Rationale for a Non-UCT Approach
>
> **TLDR:** UCT score cannot be used because of unbounded branching, while the Bayesian method can incorporate it naturally.
>
> Thank you for pointing out the need to justify the additional complexity of our Bayesian mechanism. We added the following clarification after "Adaptive Branching via the GEN Node" paragraph in Section 3.2:
>
> The UCT score is inapplicable to our AB-MCTS because GEN nodes make the problem fundamentally different from a standard multi-armed bandit problem, for which UCT was designed. In standard MCTS, the arms (branches) are static. In contrast, the GEN node in AB-MCTS dynamically generates new arms. This special problem setting, where arms are generated on the fly, prevents the direct application of UCT. We, therefore, adopt a Bayesian probabilistic model. This enables Thompson sampling based on the posterior distribution and obviates the need for complex UCB-style confidence bound analysis.
>
> ---
>
> ## [Q4-2] Comparison with Progressive Widening
>
> **TLDR:** Compared AB-MCTS with progressive widening in new experiments, showing our method is more robust and consistent.
>
> We thank the reviewer for the opportunity to clarify the key differences between our approach and progressive widening. We clarified the key differences between our approach and progressive widening in Appendix A.2 and added a new comparative experiment in Appendix C.4.
>
> ### Revision of Appendix A.2
> The progressive widening has parameters $k,\alpha$ which bounds the number of branching factors by $kn^\alpha$ with node visit count $n$. Therefore, the potential number of branches is determined a priori when we pick specific hyperparameters.
> In contrast, our approach does not limit the branching factors with hyperparameters. Instead, the branching factor adapts dynamically by the observed rewards. This is an important requirement for LLM test-time inference scaling, where a tree search that purely goes wide is known to be a strong baseline. To demonstrate the robustness of AB-MCTS, we conducted an experiment to compare AB-MCTS and progressive widening in Appendix C.4.
> ### Content of Appendix C.4
> To compare the progressive widening with AB-MCTS, we conducted an experiment on LiveCodeBench for progressive widening with various parameters. We used `deepseek-v3-0324` (the only `deepseek-v3` version available via official API at the time) with a generation budget $2^7$.
> ||$(k, \alpha)=(1, 0.45)$|$(k, \alpha)=(5, 0.5)$|$(k, \alpha)=(10, 0.55)$|AB-MCTS-A(Gaussian)|AB-MCTS-A(Beta)|AB-MCTS-M|
> |-|-|-|-|-|-|-|
> |Pass@1|48.7±0.8|50.7±1.3|50.5±3.1|48.9±2.0|**51.8±1.4**|49.6±1.6|
>
> The results are shown in Table 3. While an adequate progressive widening parameter leads to strong performance comparable to AB-MCTS, its effectiveness is highly sensitive to hyperparameters. For instance, it yields the worst result with $(k,\alpha)=(1,0.45)$. Furthermore, $(k, \alpha) = (10,0.55)$ setting shows search instability, leading to the highest variance. In contrast, AB-MCTS is robust without such tuning, demonstrating its practical advantage.
>
> ---
>
> We thank the reviewer again for their valuable feedback. We believe our responses and revisions fully address all concerns and have significantly strengthened the manuscript.

---

> ### Comment · Reviewer_Mkvu · 2025-08-05
>
> Thank you for your detailed responses! I consider W1/Q2, W2/Q3, and Q1 well addressed. I appreciate your work on getting the additional numbers.
>
> Regarding Q4-1 (UCT) and Q4-2 (progressive widening), I find the explanation still unsatisfactory. In particular, the reasoning provided for the incompatibility of UCT is the presence of the GEN node and dynamic branching, and the reasoning provided for not choosing progressive widening (PW) is that it selects the number of branches a priori based on hyperparameters. However, the latter statement is not quite accurate --- the branching factor in PW should surely be considered dynamic because $n$ varies over the course of MCTS for different nodes. The hyperparameters $k$ and $\alpha$ serve to control the *rate* of widening. Given, therefore, that PW is a valid option, one must, therefore, ask whether the introduction of GEN nodes is, in fact, necessary. With no GEN nodes, one could also then use standard UCT, which would be easily compatible with dynamic branching.
>
> My main criticism above stems from the narrative currently that adaptive branching is not possible in MCTS, whereas, as described above, one can feasibly achieve it with existing concepts.

---

> > ### Author Response · Authors · 2025-08-06
> > **[Part 1/2] Response to Official Comment by Reviewer Mkvu**
> >
> > We thank the reviewer for their detailed follow-up and for pinpointing the remaining points of unclarity. We appreciate them highlighting these important issues and for noting where our previous explanation regarding progressive widening (PW) was misleading. This helps us clarify why we believe PW is insufficient for our specific problem.
> >
> > ### The Core Distinction: PW's Reward-Agnostic vs. AB-MCTS's Reward-Sensitive Widening
> > > Regarding Q4-1 (UCT) and Q4-2 (progressive widening), I find the explanation still unsatisfactory. In particular, the reasoning provided for the incompatibility of UCT is the presence of the GEN node and dynamic branching, and the reasoning provided for not choosing progressive widening (PW) is that it selects the number of branches a priori based on hyperparameters.
> >
> > We agree with the reviewer that only the rate of widening is controlled by the hyperparameters $k$ and $\alpha$. Our intended meaning by "a priori" was that the policy for widening in PW is fixed. This policy, based only on visit counts, determines the tree's growth pattern without regard to the observed rewards discovered during the search. We have revised the explanation in Appendix A.2 to make this point clearer, as attached at the end of this response.
> >
> > > However, the latter statement is not quite accurate --- the branching factor in PW should surely be considered dynamic because $n$ varies over the course of MCTS for different nodes. The hyperparameters $k$ and $\alpha$ serve to control the rate of widening. Given, therefore, that PW is a valid option, one must, therefore, ask whether the introduction of GEN nodes is, in fact, necessary. With no GEN nodes, one could also then use standard UCT, which would be easily compatible with dynamic branching.
> >
> > > My main criticism above stems from the narrative currently that adaptive branching is not possible in MCTS, whereas, as described above, one can feasibly achieve it with existing concepts.
> >
> > We thank the reviewer for raising this important point. The central point we wish to emphasize is that the UCT score, within the PW framework, is not used to decide whether to branch or to descend to an existing child. In standard MCTS with PW, the UCT score is only used to select which child to visit, conditioned on the decision not to branch. The decision to branch is determined solely by the node's visit count, irrespective of the observed rewards.
> >
> > In contrast, AB-MCTS makes the decision of whether to branch or not based on the observed scores of the expanded nodes. This allows for a branching strategy that is truly adaptive to the observed reward landscape of the problem. We have revised Appendix A.2 to make this crucial distinction explicit. The revised text is provided at the end of this response.
> >
> > ### Node Degree Analysis of AB-MCTS and PW
> >
> > To empirically demonstrate this key difference, we analyzed the node degree distribution of AB-MCTS and PW.
> >
> > The results show AB-MCTS produces a complex, long-tailed distribution as a direct result of its reward-dependent branching. We summarize its key characteristics, as the full distribution is too varied for a simple table:
> >
> > - For AB-MCTS-M, the search favors depth: 90% of non-leaf nodes have a small degree (1-3). The remaining nodes show a long tail with degrees up to 40, which confirms that the search adaptively broadens depending on the observed rewards.
> > - AB-MCTS-A performs a much wider search. Low-degree nodes (1-3) account for 30%, and the distribution spreads broadly with degrees reaching over 100, reflecting the fact that the observed rewards affect the width of the search tree.
> >
> > In stark contrast, the node degree distribution of PW is confined to a few patterns that are highly dependent on its hyperparameters, $\alpha$ and $k$, as shown below.
> >
> > **$\alpha=0.45,k=1$**
> > |Node degree|Distribution (%)|
> > |-|-|
> > |1|55.39|
> > |2|30.92|
> > |3|4.00|
> > |4|8.31|
> > |5|0.02|
> > |9|1.37|
> >
> > **$\alpha=0.5,k=5$**
> > |Node degree|Distribution (%)|
> > |-|-|
> > |1|71.93|
> > |3|26.32|
> > |57|1.75|
> >
> > **$\alpha=0.55,k=10$**
> > |Node degree|Distribution (%)|
> > |-|-|
> > |128|100.00|
> >
> > As the tables illustrate, the node degree distribution in PW is almost entirely determined by its hyperparameters. This confirms our central argument: unlike AB-MCTS, the widening in PW does not adapt in response to the observed rewards. This reward-agnostic widening is precisely why we believe a new mechanism is necessary.
> >
> > ---

---

> > > ### Author Response · Authors · 2025-08-06
> > > **[Part 2/2] Response to Official Comment by Reviewer Mkvu**
> > >
> > > ### Revision of Appendix A.2
> > > The progressive widening has parameters $k,\alpha$ which bounds the number of branching factors by $kn^\alpha$ with node visit count $n$. With these parameters, the rule for whether to branch is pre-determined as a function of the node's visit count. Crucially, this decision does not use important information gathered during the search, namely, the observed rewards of the expanded nodes. The UCT score is only used to select which child to descend to after the decision has been made not to branch. Furthermore, defining the branching rule as a function of visit count means that the scaling behavior of the tree's shape and node degrees is pre-determined by the choice of hyperparameters.
> > >
> > > In contrast, our approach does not restrict the branching rule based solely on visit counts and hyperparameters. Instead, the branching factor adapts dynamically based on the observed rewards. This is an important requirement for LLM test-time inference scaling, where a tree search that purely goes wide is known to be a strong baseline. To demonstrate the robustness of AB-MCTS, we conducted an experiment to compare AB-MCTS and progressive widening in Appendix C.4.
> > >
> > > ---
> > >
> > > We hope this response adequately addresses the reviewer's concerns. We welcome the opportunity to provide any further clarification.

---

> > > > ### Author Response · Authors · 2025-08-08
> > > >
> > > > Dear Reviewer Mkvu,
> > > >
> > > > As the author-reviewer discussion period is approaching its end, we would like to kindly follow up on our previous response. We hope that our clarifications regarding Q4-1 (UCT) and Q4-2 (progressive widening), along with the corresponding update in Appendix A.2, have addressed your concerns. If you have any further questions, we are ready to clarify them. We hope our response is helpful to your final evaluation.
> > > >
> > > > Thank you again for your time and insightful feedback.

---

> > > > > ### Comment · Reviewer_Mkvu · 2025-08-08
> > > > >
> > > > > Thank you for all your responses and work. I've re-reviewed the paper, the discussion with other reviewers, the new explanations/clarifications provided, as well as the additional statistical evaluations and experiments. I believe the authors have provided sufficient evidence for the merit of this work, and I am happy to increase my score. I think this work is a useful contribution to the field in kindling additional ideas related to solution search with LLMs.
> > > > >
> > > > > I strongly think that the authors should dedicate a section on discussing existing solutions with a careful and dispassionate view in their final version. In the previous responses, it was said that PW does not take rewards into account, but one can argue that the visit counts are indirectly a function of the rewards - therefore, it is not entirely an accurate statement. Nevertheless, I admit that mine may be a subjective view and am willing to see it from the authors' point of view. Additionally, I would like to point to this paper [1], which I think the authors could include in their related works.
> > > > >
> > > > > [1] Monte Carlo Tree Search With Iteratively Refining State Abstractions. Sokota et al., NeurIPS 2021.

---

> > > > > > ### Author Response · Authors · 2025-08-08
> > > > > >
> > > > > > We sincerely appreciate your thorough and constructive feedback and the insightful discussion that followed. Your important questions were instrumental in sharpening the clarity of our core contribution and substantially strengthening the paper. We will be sure to incorporate your valuable final suggestions into our manuscript. Thank you again for your thoughtful engagement.

---

### Official Review · Reviewer_RYt1 · 2025-07-02

**Clarity:** 3
**Significance:** 3
**Originality:** 3
**Rating:** 5
**Confidence:** 4

**Summary:**

This paper extends MCTS-based LLM test-time computation to include progressive widening, while novelly leveraging statistical information stemming from all actions being drawn from the same LLM distribution, resulting in a potentially more efficient search. Consequently, two different approaches to utilizing this information are introduced: Adaptive-Branching MCTS Mixed-model and Adaptive-Branching MCTS Node-Aggregation. Experiments across LiveCodeBench, CodeContest, ARC-AGI, and MLE-Bench suggest that this approach is more robust in different settings at improving performance through test-time compute, compared to Best-of-N, Self-refine, and MCTS baselines.

**Questions:**

* The MCTS baselines with a branching factor of 5 end up generating 2^7-2 branches, or do you cut the last generation to 2 instead of 5? I would clarify this for completeness.
* In 4.3, is the overall width of the tree really representative of your method? Perhaps you could explore plotting the distribution of node degrees, which I believe would better show the strength of your approach.
* What do you think are the following steps to explore to try to reach the Pareto frontier? I believe it would be helpful to include a more comprehensive discussion of different designs, such as including problem difficulty and other metrics used in other works that control the relative benefit of width vs depth.

### Other suggestions

* I agree with the distinction regarding Progressive Widening in the related work. Still, given its relevance as prior work, I would encourage you to make this point more precise, potentially in the appendix.

**Ethical Concerns:**

["NO or VERY MINOR ethics concerns only"]

**Final Justification:**

I have reviewed other reviews and rebuttals, and I consider all points adequately addressed during the rebuttal.

Regarding 'Statistical Significance', I appreciate the statistical tests performed by the authors during the rebuttal, which demonstrate transparency that will be highly valuable to readers. Although the benefit of AB-MCTS is not game-changing, it indeed appears consistent. In any case, this paper provides a first and flexible step in this direction that easily admits refinements, which can be incorporated in the future to achieve stronger results in deployment settings.

Therefore, I recommend acceptance.

**Limitations:**

yes

**Paper Formatting Concerns:**

No major formatting issues were found.

**Quality:**

3

**Strengths And Weaknesses:**

## Strengths

* **Problem significance**. Test-time compute has shown great potential and is an area of great relevance and timeliness \[1\].
* **Originality/impact**. Adaptive branching could hypothetically reach the Pareto frontier for the different test-time compute methods, which seem to perform with different success for different prompts \[1\]. Consequently, the ideas proposed in this paper can be valuable to the community, and the different design choices can be progressively improved in future work to approach the Pareto frontier for varying test-time budgets.

## Weaknesses

* **Clarity.** The paper is overall well-written, but the method section lacks precision due to relying solely on text-based exposition instead of presenting the method mathematically. Since this is the core contribution of the paper, I believe a clear explanation of the method (e.g., incorporating elements from Section A.2.2, but ideally with greater clarity) would significantly enhance the paper’s quality.
  * A minor suggestion would also be to include in the appendix a walk-through example of each approach, clearly illustrating, for example, the effect of the shared parameters on the overall update for AB-MCTS-M.
* **Quality/Evaluation**. I have some doubts about the overall soundness of the empirical evidence provided by the authors.
  * **Baselines**. As far as I understand, the baselines used for MCTS (e.g., LATS \[2\] and RAP \[3\]) consider nodes as steps toward the final answer, rather than as full solutions, which differs from the formalism presented in this work. Therefore, although MCTS on solutions is a valid baseline for this paper, it misrepresents previous works. Vanilla progressive widening could also be included as a valuable ablation.
  * **Metrics**. LiveCodeBench and CodeContest are evaluated at Pass@1, while ARC-AGI is evaluated at Pass@2. I would encourage transparency and completeness to provide at least Pass@1 for ARC-AGI, too, and ideally also Pass@2 in the other tasks. Additionally, it is unclear what the metrics displayed in Tables 1 and 2 represent, especially given the inconsistency in Figures 4 and 5\.
  * **Statistical significance**. Gains in Tables 1 and 2 appear modest, especially when compared to Repeated Sampling, and it is unclear whether they are statistically significant, as no error bars are provided. In fact, confidence intervals in Figure 4 and Figure 8 in the appendix seem to suggest that gains are non-statistically significant, and the results do not evidently extend qualitatively to other LLMs.

Regarding my score, the main pain points are the “clarity” point mentioned above and the lack of transparency in the representation of other works from baselines, the metrics provided, and the statistical significance as outlined above. I am not looking for further experiments, and I don’t think modest and mixed performance improvements alone would be a reason to reject, especially given the potential long-term impact this paper could have with follow-up works that find better approaches to the Pareto frontier. Therefore, I invite the authors to address these comments and provide a clearer picture; in that case, I am willing to increase the score to ‘weak accept’. More conclusive results with more relevant improvements would be required for me to raise my score to a full ‘accept’.
—
\[1\] Snell, C., Lee, J., Xu, K., & Kumar, A. (2024). Scaling LLM Test-Time Compute Optimally can be More Effective than Scaling Model Parameters. *ArXiv*. [https://arxiv.org/abs/2408.03314](https://arxiv.org/abs/2408.03314)
\[2\] Zhou, A., Yan, K., Wang, H., & Wang, Y. (2023). Language Agent Tree Search Unifies Reasoning Acting and Planning in Language Models. ArXiv. https://arxiv.org/abs/2310.04406
\[3\] Hao, S., Gu, Y., Ma, H., Hong, J. J., Wang, Z., Wang, D. Z., & Hu, Z. (2023). Reasoning with Language Model is Planning with World Model. ArXiv. https://arxiv.org/abs/2305.14992

---

> ### Author Rebuttal · Authors · 2025-07-31
>
> We sincerely thank the reviewer for their detailed and constructive feedback. We were particularly encouraged by the reviewer's clear path to a higher score. The reviewer stated:
>
> > I invite the authors to address these comments and provide a clearer picture; in that case, I am willing to increase the score to ‘weak accept’. More conclusive results with more relevant improvements would be required for me to raise my score to a full ‘accept’.
>
> Following this valuable guidance, we have addressed every concern raised. While the rebuttal format prevents us from submitting a revised manuscript at this stage, we have already implemented all corresponding changes. We believe these revisions satisfy the conditions outlined for a 'weak accept'. Furthermore, by providing new comparative experiments and deeper analyses, we have endeavored to present the "more conclusive results and relevant improvements" that the reviewer suggested would merit a full 'accept'. We hope the reviewer will agree that our strengthened manuscript is worthy of this higher evaluation.
>
> ---
> ## [W1-1] Methodological Clarity and Formalism
> We have revised Sections 3.2, 3.3, and 3.4 to include more mathematical details and to better clarify the correspondence between each step and Algorithm 1. The revised sections are provided in Appendix 1 at the end of this response.
>
> ---
> ## [W1-2] Walk-through Examples
> We have revised our manuscript to include a detailed walk-through example section as recommended. The revised section is attached as Appendix 2 at the end of this response.
>
> ---
> ## [W2-1] Baselines
> We appreciate the reviewer's proposal for avoiding misinterpretation of the existing literature. Accordingly, we have revised the "Related Work" and "Experimental Setup" sections.
>
> We also thank them for the constructive suggestion to include the comparison with vanilla progressive widening. We added a new experiment in Appendix C.4, which evaluates the performance of progressive widening using the branching limit $kn^\alpha$ for various parameters $k,\alpha$, where $n$ is the node visit count.
> ### Content of Appendix C.4
> To compare the progressive widening with AB-MCTS, we conducted an experiment on LiveCodeBench for progressive widening with various parameters. We used `deepseek-v3-0324` (the only `deepseek-v3` version available via official API at the time) with a generation budget $2^7$.
> ||$(k, \alpha)=(1, 0.45)$|$(k, \alpha)=(5, 0.5)$|$(k, \alpha)=(10, 0.55)$|AB-MCTS-A(Gaussian)|AB-MCTS-A(Beta)|AB-MCTS-M|
> |-|-|-|-|-|-|-|
> |Pass@1|48.7±0.8|50.7±1.3|50.5±3.1|48.9±2.0|**51.8±1.4**|49.6±1.6|
>
> The results are shown in Table 3. While an adequate progressive widening parameter leads to strong performance comparable to AB-MCTS, its effectiveness is highly sensitive to hyperparameters. For instance, it yields the worst result with $(k,\alpha)=(1,0.45)$. Furthermore, $(k, \alpha) = (10,0.55)$ setting shows search instability, leading to the highest variance. In contrast, AB-MCTS is robust without such tuning, demonstrating its practical advantage.
>
> ---
> ## [W2-2] Metrics
> Our initial reporting of Pass@2 for ARC-AGI was to adhere to the standard evaluation protocol for that specific benchmark [Francois et al. 2024]. However, we agree that providing Pass@1 results improves transparency. Below are the Pass@1 results on ARC-AGI for GPT-4o:
> ||Pass@1|Pass@2|
> |-|-|-|
> |Repeated Sampling|14.0±1.7|15.0±1.0|
> |Sequential Refinement|7.7±0.6|8.7±0.95|
> |Standard MCTS|8.0±1.0|9.0±1.5|
> |AB-MCTS-M|11.0±1.0|12.3±1.2|
> |AB-MCTS-A(Gaussian)|13.0±3.6|13.0±3.6|
> |AB-MCTS-A(Beta)|12.7±0.6|14.0±2.1|
>
> The table shows consistent trends between Pass@1 and Pass@2. We observed the same trend for DeepSeek-V3. We have added these results to Appendix C.5 and revised the captions for all tables to explicitly state the metric.
>
> ---
> ## [Q1] Clarification on MCTS Baseline
> We employed the latter approach (cutting the last generation) in our experiment, and we revised our manuscript to include the clarification on this point at the end of section 4.2.
>
> ---
> ## [Q2] Further Analysis of Search Tree Topology
> We agree that analyzing the node degree distribution offers a more granular view of our algorithm's behavior and strength. We have plotted the distribution (max budget=$2^7$), and the results clearly demonstrate our method's adaptive nature:
> - For AB-MCTS-M, the search is focused on depth: 90% of non-leaf nodes have a small degree (1-3). The remaining nodes show a long tail with degrees up to 40, which confirms that the search adaptively broadens when required.
> - AB-MCTS-A performs a much wider search. Low-degree nodes (1-3) account for 30%, and the distribution spreads broadly with degrees reaching over 100.
>
> We have added these plots and a detailed discussion to Appendix C.6.
>
> ---
> ## [Q3] Towards the Pareto Frontier
> Following the advice, we have added a new discussion to the Appendix on future directions for pushing the Pareto frontier. This new section outlines two key strategies:
> 1. **Enhanced Adaptivity via Difficulty Estimation**: We propose to enhance our algorithm's existing depth-width balancing by explicitly estimating problem difficulty from collected rewards. For difficult problems (identified by low rewards), the strategy would dynamically switch, e.g., from a deep search (AB-MCTS-M) to a wide one (AB-MCTS-A). This is motivated by findings that optimal search strategies depend on difficulty [Snell et al. 2024].
> 2. **Collaborative Search with Multiple LLMs:** We also propose a method that leverages the diverse strengths of different LLMs within a single search. This is implemented by extending AB-MCTS with multiple GEN nodes, one for each LLM. Our preliminary experiment has already confirmed this to be a promising direction, showing its potential as a fruitful area for future work.
>
> ---
> ## Appendix 1: Method Revision
> ### 3.2 Adaptive Branching MCTS
> …
>
> **Thompson Sampling for Node Selection.**
> In our proposed methods, we employ a Bayesian approach with Thompson sampling for node selection. Here, we employ Thompson sampling because GEN nodes do not have child nodes, making it impossible to compute their UCT scores. In addition, Thompson sampling has the advantage of allowing node expansion in parallel. This is particularly beneficial when evaluating node scores is time-consuming, as in the case of MLE-Bench (See Appendix B.1 for MLE-Bench details).
>
> Concretely, during `SelectChild` step at line 11 of Algorithm 1, we employ Thompson sampling to decide between expanding a GEN node or selecting from existing child nodes at node $N$. Let $N$ be a node with potential actions $A_N = \\{a_0,a_1,\dots,a_{n_{\rm child}}\\}$, where the action $a_0$ corresponds to choosing the GEN node, and $a_1, \dots, a_{n_{\rm child}}$ correspond to choosing the already-existing child nodes. Suppose $P_{N}(r \mid a_i)$ is the posterior predictive distribution of the score $r$ for an eventually expanded new node ($N_{\text{new}}$ at line 5 of Algorithm 1) if we choose the action $a_i$ at node $N$. Then Thompson sampling proceeds by
> 1. Calculate $P_{N}(r \mid a_j)$ for each action $a_j$ at node $N$.
> 1. Draw scores $r_{N_\text{new},a_j}$ from $P_N(r \mid a_j)$ for each action $a_j$.
> 1. Select $\hat{a} = \arg\max_{a_j \in A_N} r_{N_\text{new},a_j}$.
>
> This three-step process corresponds to a single call to `SelectChild`.
>
> A key question is how to perform step 1, i.e., how to model and calculate $P_N(r \mid a_j)$ for all $a_j$, in particular for $j=0$ (i.e., GEN node). …
>
> (3.3 and 3.4 revisions omitted due to length limit)
>
> ---
> ## Appendix 2: Walk‑through Examples
> We walk through an iteration of AB-MCTS-M and AB-MCTS-A on the example trees in Figures 2 and 3. The process is stochastic due to Thompson sampling; for clarity, we assume specific sampled outcomes.
>
> ### AB‑MCTS‑M
> AB-MCTS-M incrementally builds the search tree by adding one node at a time. This section details one complete iteration of the AB-MCTS-M algorithm, clearly illustrating the sequence of selecting a node to expand, performing the expansion, and backing up the score. For simplicity and concreteness, we use the tree structure depicted in Figure 2.
>
> 1. **($N \to N_1$)** At $N$, we compute posterior distributions for its four children, GEN, $N_1$, $N_2$, and $N_3$ under the mixed model, and draw one score from each. If GEN receives the highest sample, it is expanded and its score is backed up (Algorithm 1, lines 11–13). Here, we assume that $N_1$ attains the highest sampled score, reflecting exploitation of a child whose posterior peak is comparatively large.
> 2. **($N_1 \to N_1'$)** $N_1$ has two direct children, $N_1'\;(r = 0.8)$ and $N_2'\;(r = 1.0)$. We compute posteriors for GEN, $N_1'$, and $N_2'$, then sample again. Although the subtree $T(N_2')$ currently contains nodes with higher score, as we can see from Figure 2, the finite variance of its posterior ensures that $N_2'$ is not always selected, and encourages more exploration for under-explored tree regions. Suppose $N_1'$ is chosen on this iteration.
> 3. **(Expanding $N_1'$)** Because $N_1'$ is a leaf, we expand it (Algorithm 1, line 10). Assume the newly generated node receives the score $r = 0.5$. Since all the leaf nodes have a GEN child, a GEN node is appended to this expanded node as well.
> 4. **(Score backup)** The score is backed up to the ancestors, and the sampled score influences subsequent posterior updates. At the next `SelectExpansionTarget` call, the four posteriors at $N$ have different shapes; specifically, the peak of $N_1$’s posterior shifts left due to the lowered expected value. Other posterior distributions are affected as well, e.g., the right‑hand tail of the GEN posterior contracts. Please note that, in AB-MCTS-M, the change of posterior distribution shape cannot be analytically written down and is calculated by MCMC. Thus, unlike standard MCTS, the score backup step involves appending the new score to a list of scores.
>
> (AB-MCTS-A explanation omitted due to length limit)

---

> > ### Comment · Reviewer_RYt1 · 2025-08-02
> >
> > I appreciate the author’s work performed during the rebuttal period and their thoughtful rebuttal comments.
> >
> > **W2-2** I appreciate the clarification on the metrics. It would be essential to clarify that some columns represent Pass@1 while others represent Pass@2 in Tables 1 and 2, but I now consider the point resolved.
> >
> > **W2-1** Thank you for clarifying the comparison with other works to avoid their misrepresentation. Could you provide an excerpt of the additions to understand how this will be specifically addressed?
> >
> > **Q2** I am also happy to see the encouraging results on the tree topology, and the fact that there is significant flexibility observed, which validates the methods work as expected.
> >
> > **W1-1** In terms of clarity, I appreciate the revisions, and from the excerpt provided, the changes go a long way in the right direction. I am especially intrigued about sections 3.3 and 3.4, as the mechanism for propagating new evidence through the trees was the part that was least clear to me in both versions of the method. Similarly, the walk-through example is looking great. Still, I would appreciate a more in-depth discussion of the _Score backup_ step, as well as how the posterior distributions for a GEN node and one of the children look in step 1.
> >
> > **W1-3** Finally, I’d like to highlight a point that was not addressed in the rebuttal comment.
> >
> > > **Statistical significance**. Gains in Tables 1 and 2 appear modest, especially when compared to Repeated Sampling, and it is unclear whether they are statistically significant, as no error bars are provided. In fact, confidence intervals in Figure 4 and Figure 8 in the appendix seem to suggest that gains are non-statistically significant, and the results do not evidently extend qualitatively to other LLMs.
> >
> > Upon reviewing the rebuttal to other reviews, I am more strongly convinced that the improvements are not substantial and, in most cases, not statistically significant, which will still prevent me from raising the score to a full 'accept'.

---

> > > ### Author Response · Authors · 2025-08-03
> > > **[Part 1/4] Response to Official Comment by Reviewer RYt1**
> > >
> > > We thank the reviewer for the prompt and constructive feedback. We are pleased to see that our rebuttal has fully addressed the concerns regarding W2-2 and Q2, and partially addressed those for W2-1 and W1-1.
> > >
> > > Please be aware that the previous rebuttal was constrained by a 10,000-character limit. This made it difficult to elaborate on all points as thoroughly as we would have liked. We appreciate this opportunity to provide further details below.
> > >
> > > ---
> > >
> > > ## [W1-1] Methodological Clarity and Formalism
> > > We thank the reviewer for pointing out the need for further clarification on Sections 3.3 and 3.4. Due to space constraints, we could not include these sections in our previous rebuttal. We have provided the revised sections in Appendix 3 at the end of this response, demonstrating the correspondence between mathematical formulations and the algorithmic steps.
> > >
> > > Regarding the posterior distributions of GEN nodes, the right side of Figure 2 illustrates the posterior calculated by MCMC for node $N$ in the tree structure shown on the left side of Figure 2. We modified the manuscript to explicitly clarify this point within the description of Figure 2. Concerning changes in posterior distributions upon expanding a new node, we are currently unable to include visualizations here due to restrictions on PDF attachments. However, as described in our previous response, the walkthrough examples provide an explanation of these posterior updates for an example tree in Figure 2.
> > >
> > > Additionally, we thank the reviewer for highlighting the need to further improve the clarity of our explanation regarding the score backup mechanism. To address this, we have revised the walkthrough explanation as shown in Appendix 4 at the end of this response and will refer to this updated description from sections 3.3 and 3.4. Furthermore, in Appendix 4, we added a walkthrough example for AB-MCTS-A, which was previously excluded due to the length limit.
> > >
> > > ---
> > >
> > > ## [W2-1] Baselines
> > >
> > > We thank the reviewer for their follow-up and for acknowledging our clarification on W2-1. We are happy to provide the specific excerpts from our revised manuscript to show exactly how this has been addressed.
> > >
> > > Below are the changes made to the 'Related Work' and 'Experiments' sections.
> > >
> > > ### Revisions in 'Related Work'
> > >
> > > To more accurately position our work, we have removed lines 85-89 and replaced them with the following paragraph:
> > >
> > > > LATS [9], RAP [15], SWE-Search [10], and RepoUnderstander [11] combine LLMs with MCTS, primarily targeting sequential decision making. In this context, nodes represent states and edges represent actions, which may involve interaction with an environment. LATS utilizes API calls and code execution as actions to solve tasks. RAP addresses the process of solving block-moving puzzles and mathematical word problems step-by-step. SWE-Search explores sequences of actions such as searching, editing, and running tests to resolve issues within a software repository. RepoUnderstander employs MCTS for exploration on a repository knowledge graph. The application of LATS to coding tasks [9, Section 5.2] aligns with the context of multiple-answer generation in this paper and corresponds to what we refer to as “standard MCTS” in our experiments.
> > >
> > > ### Revisions in 'Experiments'
> > >
> > > Correspondingly, we have made the following updates to the 'Experiments' section to ensure consistency:
> > >
> > > 1. The citation to RAP [15] has been removed from Table 1.
> > > 2. The description of Standard MCTS (formerly lines 245-249) has been edited as follows:
> > > > (3) Standard MCTS follows the configuration from LATS [9, Section 5.2]. Each expansion adds five child nodes, and the search proceeds until it reaches the 2^7 nodes
> > >
> > > ---

---

> > > > ### Author Response · Authors · 2025-08-03
> > > > **[Part 2/4] Response to Official Comment by Reviewer RYt1**
> > > >
> > > > ## [W1-3] Statistical Significance
> > > >
> > > > We sincerely thank the reviewer for raising the important point regarding statistical significance again. Due to the character limit in our initial rebuttal, a detailed explanation of this point could not be included in the previous response. To improve the transparency of our results, we have added error bars to our result tables.
> > > >
> > > > Regarding the comment that the performance gains appear modest, we would like to argue that the primary strength of our method, AB-MCTS, lies in its robust and stable performance across a wide range of benchmarks and LLMs.
> > > >
> > > > For instance, while repeated sampling excels on ARC-AGI, it is not superior to other methods on LiveCodeBench, CodeContests, or the Nomad2018 and Spooky of MLE-Bench. Conversely, sequential refinement performs very well on Nomad2018 (MLE-Bench) but struggles on ARC-AGI and CodeContests. Similarly, standard MCTS achieves the highest performance with DeepSeek-V3 on LiveCodeBench, but its performance is inconsistent across other combinations.
> > > >
> > > > In contrast, as shown in Tables 1 and 2, AB-MCTS consistently achieves a high rank regardless of the specific benchmark or LLM used. We believe this robustness is a highly valuable trait for LLM inference scaling. Indeed, the high cost of these experiments (often several hundred dollars for a single condition) makes it practically difficult to run the sheer volume of trials needed to verify the statistical significance of every performance gain. Given these conditions, a method like AB-MCTS that offers predictably strong performance serves as a compelling and reliable first choice.
> > > >
> > > > To statistically validate this claim of robustness, we conducted a statistical analysis of the rankings for AB-MCTS-M and the three other baseline methods across all benchmarks and LLMs.
> > > >
> > > > The mean ranks and standard deviations are as follows:
> > > > - AB-MCTS-M: 1.44 ± 0.52
> > > > - Repeated Sampling: 2.44 ± 1.13
> > > > - Sequential Refinement: 3.11 ± 1.05
> > > > - Repeated Sampling: 2.88 ± 1.05
> > > >
> > > > First, a Friedman test was conducted, which yielded a p-value of $p=0.0251$. As this is less than $0.05$, it confirms that a statistically significant difference exists among the group of methods.
> > > >
> > > > Next, we conducted post-hoc Wilcoxon signed-rank tests to compare AB-MCTS-M against each of the three baselines. The resulting p-values are:
> > > > - AB-MCTS-M vs. Repeated Sampling, p=0.043
> > > > - AB-MCTS-M vs. Sequential Refinement, p=0.0078
> > > > - AB-MCTS-M vs. Standard MCTS, p=0.0117
> > > >
> > > > After applying the Bonferroni correction ($\alpha=0.05/3≈0.0167$), AB-MCTS-M demonstrates a statistically significant advantage over both sequential refinement ($p=0.0078$) and standard MCTS ($p=0.0117$).　We acknowledge that its advantage over repeated sampling does not meet this strict criterion. However, we argue that the clear overall trend, evidenced by AB-MCTS-M having the lowest mean rank and the smallest standard deviation, strongly indicates its superior robustness.
> > > >
> > > > We hope that this new analysis adequately addresses the reviewer's concerns regarding the significance of our results.
> > > >
> > > > ---

---

> > > > > ### Author Response · Authors · 2025-08-03
> > > > > **[Part 3/4] Response to Official Comment by Reviewer RYt1**
> > > > >
> > > > > ## Appendix 3: Method Revision
> > > > >
> > > > > ### 3.3 AB-MCTS-M: Adaptive Branching MCTS with Mixed Model
> > > > >
> > > > > To model $P_N(r \mid a_j)$, we employ a node-specific mixed model fitted individually at each node $N$. That is, we fit a separate model for each $N$ every time `SelectChild` in Algorithm 1 is invoked.
> > > > > Denoting $r_{N_{\text{new}},a_j} \sim P_N(r \mid a_j)$ as a score of an eventually expanded node $N_{\text{new}}$ if we choose an action $a_j$ at $N$, our mixed model is given as:
> > > > > $$\begin{gathered}
> > > > >     r_{N_{\text{new}}, a_j} = \alpha_j + \sigma_y\epsilon_{N_{\text{new}}}, \quad
> > > > >     \alpha_j = \mu_{\alpha} + \sigma_{\alpha}\epsilon_j, \\
> > > > >     \epsilon_{N_\text{new}} \sim \mathcal{N}(0, 1), \quad \epsilon_j \sim \mathcal{N}(0, 1),
> > > > > \end{gathered}$$
> > > > > Here, $\alpha_j$ is a "group-level" intercept capturing the quality of the base solution at $N_j$, while $\sigma_y\epsilon_{N_\text{new}}$ represents per-instance noise.
> > > > > The GEN node (action $a_0$) is treated as a newly introduced group without its own direct observations. However, its group-level intercept $\alpha_0$ is inferred not from the prior alone but rather from the posterior distribution over $\mu_{\alpha}$ and $\sigma_{\alpha}$, which is informed by the other observed data.
> > > > > We assume that even after multiple refinement stages, the quality associated with the answer at node $N_j$ continues to be captured by this shared parameter (see Appendix A.2.2 for further details).
> > > > >
> > > > > **Algorithm Outline.**
> > > > > To model $P_N(r \mid a_j)$, AB-MCTS-M assigns each subtree under $N_j$, denoted as $T_{\text{sub}}(N_j)$, as a distinct group $j$ (see Figure 2 for example subtree).
> > > > > The mixed model leverages observed scores from these groups to compute the posterior predictive distributions of expected scores for new nodes generated from each group (See Figure 2 for a schematic illustration).
> > > > > We sample the scores from all the groups (the GEN node and $T_{\text{sub}}(N_j)$) using calculated posterior predictives. If the GEN node's sampled score is highest, we call $f_\text{LLM}$ to generate a new child node. Otherwise, we choose the child node $N_j$ with the highest score and continue the sampling step.
> > > > >
> > > > > **Score Backup Mechanism.**
> > > > > When a new node $N$ is created, its observed score is added to the histories of $N$ and its ancestors. This cumulative record is used to update the posterior distributions in the mixed model. The observed score is not backed up to a GEN node, but it indirectly influences the GEN node's score probability distribution through the shared parameters in the mixed model (see Appendix A.3 for a detailed walkthrough).
> > > > >
> > > > > ### 3.4 AB-MCTS-A: Adaptive Branching MCTS with Node Aggregation
> > > > >
> > > > > ...
> > > > >
> > > > > **Algorithm Outline.**
> > > > > AB-MCTS-A aggregates all child nodes under a single CONT node, which represents refinements from existing child nodes (see Figure 3). We model each node's score probability in a Bayesian framework and perform Thompson sampling on posterior predictives to decide between generating a new child (GEN) or refining an existing one (CONT). In contrast to AB-MCTS-M, we do not use shared parameters among different node probability distributions.
> > > > >
> > > > > To model $P_N(r \mid a_j)$, we utilize exponential family distributions with conjugate priors, enabling analytical and efficient posterior updates. We employ two variants:
> > > > > 1. AB-MCTS-A (Gaussian), using a normal-inverse-$\chi^2$ prior for unbounded scores: $P_N(r \mid a_j) = p(r \mid \{r_k\}_{k=1}^K) = \mathcal{N}(\mu \mid \hat{m}, \tfrac{\sigma^2}{\hat{\kappa}})\chi^{-2}(\sigma^2 \mid \hat{\nu}, \hat{\tau}^2)$, and
> > > > > 1. AB-MCTS-A (Beta), using a Beta prior for scores in $[0,1]$: $P_N(r \mid a_j) = p(r\mid \{r_k\}_{k=1}^K) = B(r \mid \hat{\alpha}, \hat{\beta})$,
> > > > >
> > > > > where $r_k$ represents the scores backed up to the node $N_j$ (GEN node, CONT node or LLM-generated chlid nodes; see Figure 3 for example tree), where $\hat{m}, \hat{\kappa},\hat{\nu},\hat{\tau},\hat{\alpha}, \hat{\beta}$ are determined from observed scores $r_k$ and updated as these scores are backed up. The detailed parameter update rules are given in Appendix A.3.
> > > > >
> > > > > **Score Backup Mechanism.**
> > > > > During score-backup operations, the expanded node score is backed up to the GEN node which led to the expansion of that node and the GEN node's ancestors  (see Appendix A.3 for a detailed walkthrough). As we can see from Figure 3, a GEN node's ancestors include only generated nodes and CONT nodes, so the score is backed up to a GEN node only from the node that is created by choosing that GEN node. The backed-up score is used to update prior probability distribution parameters.
> > > > >
> > > > > ---

---

> > > > > > ### Author Response · Authors · 2025-08-03
> > > > > > **[Part 4/4] Response to Official Comment by Reviewer RYt1**
> > > > > >
> > > > > > ## Appendix 4: Walk‑through Examples
> > > > > >
> > > > > > ### AB-MCTS-M
> > > > > > ...
> > > > > >
> > > > > > 4. **(Score backup)** The score is propagated upwards from the expanded node toward the root, as in standard MCTS. In the current example, the score is backed up through the node generated at step 3, then through nodes $N_1’$, $N_1$, and $N$. Unlike standard MCTS, AB-MCTS-M maintains individual scores rather than averages. Specifically, the backed-up scores for nodes $N_1, N_2, N_3$ form distinct observation lists corresponding to each group in the mixed model. Although GEN nodes have no direct observations due to this score backup rule, their posterior distributions share statistical strength with these groups, allowing indirect information sharing and improved estimation accuracy for a score obtained by node expansion. …
> > > > > >
> > > > > > ### AB‑MCTS‑A
> > > > > > AB-MCTS-A works in a similar manner to AB-MCTS-M, except for the introduction of CONT nodes and how we perform score backup. To clearly illustrate the algorithm and highlight the difference from AB-MCTS-M, we describe a complete iteration of selection, expansion, and score backup cycle for AB-MCTS-A. Since the only difference between Beta and Gaussian variants is how the score update is reflected in the posterior distribution, here we focus on the qualitative aspect of posterior update and focus on the details for an example tree, Figure 3.
> > > > > >
> > > > > > 1. **($N \to \text{CONT}$)** At the root, we sample from the posteriors of the GEN and CONT children. As detailed in Section 3.4, the GEN posterior is informed by the CONT children nodes $N_1\;(0.8)$, $N_2\;(0.0)$, $N_3\;(0.2)$, whereas the CONT posterior uses the CONT node's descendant scores excluding $N_1$, $N_2$ and $N_3$, i.e., 0.8, 1.0, 0.3. We assume CONT is selected here.
> > > > > > 2. **($\text{CONT} \to N_1$)** Next we compute posteriors for CONT's children $N_1$, $N_2$, and $N_3$. Due to the score-backup rule, the posterior distributions are computed from previously expanded nodes; Concretely, the following scores are used for posterior distribution calculation: $N_1$: (0.8, 0.8, 1.0), $N_2$: (0.0), and $N_3$: (0.2, 0.3). Here, we assume $N_1$ obtains the highest sample via Thompson sampling.
> > > > > > 3. **(Expanding $N_1$)** Again, we perform Thompson sampling between the GEN and CONT children of $N_1$. The GEN posterior uses (0.8, 1.0); the CONT posterior falls back to the prior because no generated descendants exist. Here, we assume GEN is selected, leading to the expansion of a new node under $N_1$’s CONT child. We assume the score $r = 0.5$.
> > > > > > 4. **(Score backup)** We backup the score $r = 0.5$ as prescribed in Section 3.4: First, the score is backed up to (i) the GEN node which generates the node. Second, it propagates to: (ii) the GEN node's ancestor $N_1$, and  (iii) the CONT node that is $N_1$’s parent. Because 0.5 is lower than the existing scores 0.8 and 1.0, the posterior peak of $N_1$ shifts left, reducing the probability that $N_1$ will be chosen again from CONT. Similarly, adding 0.5 to the existing scores (0.8, 1.0, 0.3) lowers the peak of the CONT posterior at $N$, thus decreasing the probability that CONT will be selected at $N$ in later iterations.
> > > > > >
> > > > > > ---
> > > > > >
> > > > > > Thank you again for the opportunity to provide additional clarification. We believe this response addresses your remaining concerns.

---

> > > > > > > ### Comment · Reviewer_RYt1 · 2025-08-03
> > > > > > >
> > > > > > > Thank you for your thorough and on-point changes to address the points initially raised; I believe all of them lead to substantial improvements to the paper.
> > > > > > >
> > > > > > > I consider all of my points adequately addressed. Regarding Statistical Significance, I appreciate the statistical tests performed by the authors during the rebuttal, which demonstrate transparency that will be highly valuable to readers. Although the benefit of AB-MCTS is not game-changing, it indeed appears consistent. In any case, this paper provides a first and flexible step in this direction that easily admits refinements, which can be incorporated in the future to achieve stronger results in deployment settings.
> > > > > > >
> > > > > > > Consequently, I will raise my score to 'accept'.

---

> > > > > > > > ### Author Response · Authors · 2025-08-05
> > > > > > > >
> > > > > > > > We sincerely appreciate the reviewer’s thorough and constructive feedback, which greatly enhanced both the clarity and rigor of our manuscript. Your insightful suggestions have substantially strengthened our paper. Thank you again for your detailed and thoughtful review.

---

### Decision · Program_Chairs · 2025-09-17

**Decision:**

Accept (spotlight)

**Comment:**

This paper extends MCTS-based test-time computation for LLMs by incorporating progressive widening, while leveraging the statistical property that all actions are drawn from the same LLM distribution. This enables a more efficient search process. Building on this idea, the authors propose two approaches: Adaptive-Branching MCTS Mixed-Model and Adaptive-Branching MCTS Node-Aggregation. Experiments on LiveCodeBench, CodeContest, ARC-AGI, and MLE-Bench demonstrate that the proposed methods are more robust across diverse settings, outperforming Best-of-N, Self-Refine, and standard MCTS baselines in improving performance through test-time computation.

The studied problem of this paper is very important, and the experiments are comprehensive. There are several minor weaknesses in the experiments as mentioned by the reviewers, and the authors have provided sufficient feedback.

In the rebuttal, the authors have addressed all the concerns proposed by the reviewers. The final scores from all the reviewers are positive. I recommend acceptance.